# Latitudinal variations of $\delta^{30}$Si and $\delta^{15}$N signatures along the Peruvian shelf: quantifying the effects of nutrient utilization versus denitrification over the past 600 years

Kristin Doering[1*,2], Claudia Ehlert[3], Philippe Martinez[4], Martin Frank[2], Ralph Schneider[5]

[1*]now at: Department of Oceanography, Dalhousie University, Halifax, Nova Scotia, Canada
[2]GEOMAR Helmholtz Centre for Ocean Research Kiel, Kiel, 24148, Germany
[3]Max Planck Research Group - Marine Isotope Geochemistry, Carl von Ossietzky University, 26129 Oldenburg, Germany
[4]Université de Bordeaux, CNRS, Environnements et Paléoenvironnements Océaniques et Continentaux (EPOC), UMR 5805, Pessac, France
[5]Institute of Geosciences, University of Kiel, Kiel, 24118, Germany

*Correspondence to*: Kristin Doering (Kristin.Doering@dal.ca)

## Abstract

The sedimentary stable nitrogen isotope compositions of bulk organic matter ($\delta^{15}$N$_{bulk}$) and silicon isotope composition of diatoms ($\delta^{30}$Si$_{BSi}$) both mainly reflect the degree of past nutrient utilization by primary producers. However, in ocean areas where anoxic and suboxic conditions prevail, the $\delta^{15}$N$_{bulk}$ signal ultimately recorded within the sediments is also influenced by water column denitrification causing an increase in the subsurface $\delta^{15}$N signature of dissolved nitrate ($\delta^{15}$NO$_3^-$) upwelled to the surface. Such conditions are found in the oxygen minimum zone off Peru, where at present an increase in subsurface $\delta^{15}$NO$_3^-$ from North to South along the shelf is observed due to ongoing denitrification within the pole-ward flowing subsurface waters, while the $\delta^{30}$Si signature of silicic acid ($\delta^{30}$Si(OH)$_4$) at the same time remains unchanged.

Here, we present three new $\delta^{30}$Si$_{BSi}$ records between 11°S and 15°S and compare these to previously published $\delta^{30}$Si$_{BSi}$ and $\delta^{15}$N$_{bulk}$ records from Peru covering the past 600 years. We present a new approach to calculate past subsurface $\delta^{15}$NO$_3^-$ signatures based on the direct comparison of $\delta^{30}$Si$_{BSi}$ and $\delta^{15}$N$_{bulk}$ signatures at a latitudinal resolution for different time periods. Our results show that, during the Current Warm Period (CWP, since 1800 AD) and prior short-term arid events, source water $\delta^{15}$NO$_3^-$ compositions have been close to modern values increasing southward from 7 to 10‰ (between 11°S and 15°S). In contrast, during the Little Ice Age (LIA) we calculate low $\delta^{15}$NO$_3^-$ values between 6 and 7.5‰. Furthermore, the direct $\delta^{30}$Si$_{BSi}$ versus $\delta^{15}$N$_{bulk}$ comparison also enables us to relate the short-term variability in both isotope compositions to changes in the ratio of nutrients (NO$_3^-$: Si(OH)$_4$) taken up by different dominating phytoplankton groups (diatoms and non-siliceous phytoplankton) under the variable climatic conditions of the past 600 years. Accordingly, we estimate a shift from a 1:1 (or 1:2) ratio during the CWP and a 2:1 (up to 15:1) ratio during the LIA, associated with a shift from overall high nutrient utilization to NO$_3^-$ dominated (and thus non-siliceous phytoplankton) utilization.

## 1. Introduction

Investigations of the isotopic compositions of the macro-nutrients, such as silicic acid ($Si(OH)_4$) and nitrate ($NO_3^-$), have been used to infer changes of biogeochemical cycles in the past (Brunelle et al., 2007; Horn et al., 2011; Robinson et al., 2014). The preferential incorporation of the lighter isotopes $^{14}N$ and $^{28}Si$ into organic matter (OM) and biogenic opal (BSi), respectively, during primary production in surface waters leads to an increase in the $\delta^{15}N$ and $\delta^{30}Si$ in the remaining dissolved nutrients (i.e. $\delta^{15}NO_3^-$ and $\delta^{30}Si(OH)_4$) as a result of progressive consumption of the nutrient pools (Altabet et al., 1991; De La Rocha et al., 1997; Wada and Hattori, 1978). This preferential incorporation is associated with an approximate enrichment factor of -5‰ for $NO_3^-$ (Waser et al., 1998) and -1.1‰ for $Si(OH)_4$ (De La Rocha et al., 1997), which agree well with estimates for the Peruvian shelf (Ehlert et al., 2012; Mollier-Vogel et al., 2012; Grasse et al., 2016). While a potential fractionation of $\delta^{30}Si$ signatures of biogenic opal during dissolution of -0.55‰ has been reported previously (Demarest 2009), investigations from the water column of the Southern Ocean did not find significant difference between the $\delta^{30}Si$ values of particles in the water column and in surface sediments (Varela et a., 2004; Fripiat et al., 2012; Closset et al., 2015). Furthermore, field studies and laboratory experiments based on sediments have so far indicated that $\delta^{30}Si$ signatures of diatoms within the sediments are generally unaffected by diagenetic alteration (e.g. Egan et al., 2012; Wetzel et al., 2014; Ehlert et al., 2016). Accordingly, the degree of utilization of $NO_3^-$ and $Si(OH)_4$ is recorded in the $\delta^{15}N_{bulk}$ and $\delta^{30}Si_{BSi}$ of the OM and BSi produced. In combination with parameters such as organic carbon, BSi or barium accumulation rates, both $\delta^{15}N_{bulk}$ and $\delta^{30}Si_{BSi}$ have been employed as proxies for the evaluation of past productivity and corresponding nutrient utilization (De La Rocha et al., 1998; François et al., 1992; Horn et al., 2011; Pichevin et al., 2005).

However, in coastal upwelling areas, where upwelling of nutrient-rich subsurface waters causes high surface productivity, subsequent degradation of the high amounts of OM leads to extensive oxygen consumption in the water column (Pennington et al., 2006; Zuta and Guillén, 1970). As a result of the low oxygen concentrations, $NO_3^-$ is used as an oxidant during OM degradation and is transferred to $N_2$ leading to a net loss of bio-available nitrogen (e.g., denitrification and anaerobe ammonium-oxidation; (Codispoti, 2006; Lam et al., 2009). Due to the high isotope fractionation factor (~ -20‰) associated with denitrification, the $\delta^{15}NO_3^-$ signatures of subsurface waters strongly increase and consequently supply a heavy $\delta^{15}NO_3^-$ signal to surface waters during upwelling (Cline and Kaplan, 1975). This $^{15}N$-enriched $NO_3^-$ is incorporated by phytoplankton and ultimately deposited and buried in marine sediments. Accordingly, although $\delta^{15}N_{bulk}$ also varied in phase with productivity proxies, elevated $\delta^{15}N_{bulk}$ values in highly productive and poorly ventilated regions including most of the coastal upwelling areas, have been generally interpreted as the consequence of stronger denitrification associated with intense oxygen depletion (Agnihotri et al., 2006; 2008; De Pol-Holz et al., 2007; Fleury et al., 2015; Gutiérrez et al., 2009; Mollier-Vogel et al., 2012; Salvatteci et al., 2014b). However, given that dissolved $\delta^{15}NO_3^-$ is influenced by both nutrient utilization and denitrification – associated with water column de-oxygenation - both processes should also influence the $\delta^{15}N_{bulk}$ signatures recorded by the sedimentary OM.

In contrast, $\delta^{30}Si_{BSi}$ signatures are primarily controlled by surface water diatom productivity and $Si(OH)_4$ utilization (Brzezinski, 2002; De La Rocha et al., 1998) closely coupled to the amount of

upwelling strength in the study area (Doering et al., 2016; Ehlert et al., 2013; 2015; 2012; Grasse et al., 2013). Accordingly, downcore records of $\delta^{30}Si_{BSi}$ off Peru are closely coupled to changes in the diatom assemblage with high signatures (>1‰) reflecting strong upwelling conditions and lower signatures (0.5-1‰) reflecting weak upwelling conditions (Doering et al., 2016). This coupling was previously shown to be mainly the consequence of changes in the relative abundance of different diatom groups during diatom succession linked to different upwelling strength (Doering et al., 2016) rather than potential species-specific fractionation (Sutton et al., 2013).

Thus, the combination of both $\delta^{30}Si_{BSi}$ and $\delta^{15}N_{bulk}$ compositions in the water column and late Quaternary sediments off Peru has been applied as a measure to disentangle modern and past nutrient utilization and denitrification processes (Ehlert et al., 2015; Grasse et al., 2016). Comparison of modern dissolved $Si(OH)_4$ and $NO_3^-$ distributions and their corresponding isotopic ratios has shown that $Si(OH)_4$ and $NO_3^-$ concentrations and their stable isotopic signatures are strongly correlated within the surface mixed layer at near-shore and offshore areas, indicating that the signal preserved in the sediments depends on the degree of utilization of both nutrients (Grasse et al., 2016). Similarly, an initial comparison for the past 600 years based on one sediment core indicated that both isotope compositions were largely influenced by nutrient utilization suggesting that denitrification in the water column only had a significant influence since ~1850 AD (Ehlert et al., 2015) thus partly contradicting previous interpretations of N-loss having been the main driver of changes in past $\delta^{15}N_{bulk}$ records.

At present the features of the Peruvian upwelling system vary significantly with El Niño-Southern Oscillation (ENSO) on interannual time scales. During warm phases of ENSO (El Niño) a weakening of the trade winds over the equatorial Pacific and an eastward displacement of West Pacific warm pool (Picaut et al., 1996) cause warmer sea surface temperature anomalies in the Central and Eastern Pacific Ocean. Off Peru this causes the mixed layer (and thermocline/nutricline) depth to increase, decreasing the nutrient content ($NO_3^-$ and iron (Fe)) of upwelled waters and leading to a decrease in phytoplankton (mainly diatoms) abundance and productivity (Barber and Chávez, 1983; Chavez, 1989; Espinoza-Morriberón et al., 2017; Sanchez et al., 2000). In contrast, the cold phases of ENSO (La Niña) are associated with a stronger Walker circulation (west-east or zonal) and upwelling-favorable winds off Peru, resulting in negative sea surface temperature (SST) anomalies (Morón, 2000), a thermocline shoaling and higher phytoplankton productivity (Espinoza-Morriberón et al., 2017). Similar conditions have been reported to alternate on the multicentennial time scales during Northern Hemisphere cold and warm periods. These so-called El Niño- and La Niña-like mean states reflect larger-scale oceanographic and climatic changes (Fleury et al., 2015; Rein, 2004; Yan et al., 2011). Accordingly, the climate of the last 600 years can be divided into two climatic phases consisting of the Current Warm Period (CWP, since 1800 AD) and the Little Ice Age (LIA, ca. 1400 to 1800 AD). Off Peru, the CWP has been characterized by dry (arid) conditions, strong upwelling intensity, as well as high productivity and intense N-loss processes, reflecting overall dominant La-Niña conditions (Fleury et al., 2015; Salvatteci et al., 2014b; Sifeddine et al., 2008). In contrast, within the present day main upwelling area between 10°S and 15°S, the LIA was characterized by lower productivity and low denitrification intensity for the present day main upwelling area between 10°S and 15°S (Díaz-Ochoa et al., 2009; Salvatteci et al., 2014b; Sifeddine et al., 2008). Previous paleo-reconstructions agreed that these conditions were induced by weakening of the Walker circulation and reduction of the South Pacific

subtropical High (SPSH), as well as by a southward shift of the mean position of the Intertropical Convergence Zone (ITCZ) and the associated precipitation belt (Fleury et al., 2015; Sachs et al., 2009; Salvatteci et al., 2014b; Sifeddine et al., 2008). These changes resulted in reduced LIA surface productivity and more oxygenated subsurface waters off Peru, as reflected by lower sedimentary BSi and TOC concentrations (Ehlert et al., 2015; Gutiérrez et al., 2009; Salvatteci et al., 2014a) and Si/Fe ratios (Fleury et al., 2015), and supported by a marked reduction in the sedimentary concentrations of redox sensitive trace metals such as molybdenum and rhenium (Salvatteci et al., 2014b; Sifeddine et al., 2008). However, these conditions did not prevail continuously but instead short-term variations during both the LIA and the CWP are for example mirrored by changes in diatom abundances, productivity sensitive element ratios indicative of productivity changes (Br/Fe) and $\delta^{15}N_{bulk}$ values (Fleury et al., 2015). These proxy records indicate multidecadal shifts between arid/humid conditions during the CWP and particularly during the LIA when pronounced short-term periods of arid conditions occurred (Fleury et al., 2015). The well-studied biogeochemical evolution of the Peruvian shelf over the last 600 years and the significant differences in productivity and subsurface oxygenation between the CWP and the LIA form the basis for our study to gain new insights into the relationship between nutrient utilization and denitrification via $\delta^{30}Si_{BSi}$ and $\delta^{15}N_{bulk}$ records.

Here, our goal is to verify whether the southward increase of $\delta^{15}NO_3^-$ due to denitrification observed in the present day has persisted during the marked changes in upwelling intensity during the LIA and CWP, and therefore under different ENSO conditions, based on comparison of $\delta^{30}Si_{BSi}$ and $\delta^{15}N_{bulk}$ signatures of four different sediment cores retrieved along the entire gradient of upwelling strength of the southern Peruvian shelf. More specifically, we aim to detect the extent of variability in $\delta^{15}N_{bulk}$ caused as a function of denitrification and nutrient utilization during specific time periods (i.e. LIA and CWP). Therefore, we present three new records for $\delta^{30}Si_{BSi}$ and BSi concentrations from the Peruvian shelf between 11°S and 15°S covering the last 600 years. These are compared to previously published $\delta^{15}N_{bulk}$ data obtained from the same cores (Fleury et al., 2015) and $\delta^{30}Si_{BSi}$ and $\delta^{15}N_{bulk}$ records from a fourth core from 14°S (Fig.1; Ehlert et al., 2015).

**Regional Setting**

Along the Peruvian margin the main source for the high amounts of upwelled nutrients (30 µmol $L^{-1}$ for both $Si(OH)_4$ and $NO_3^-$; Bruland et al., 2005) is the subsurface Peru-Chile Undercurrent (PCUC), which flows southward along the continental slope and outer shelf between 4°S and 14°S at a depth between 50 and 150 m, before it detaches from the shelf south of 15°S (Brink et al., 1983; Chaigneau et al., 2013; Toggweiler et al., 1991). Eastward flowing subsurface waters of the Equatorial Undercurrent (EUC) and the Southern Subsurface Counter Current (SSCC) (see Fig. 1) feed the PCUC. These subsurface currents deliver $Si(OH)_4$ and $NO_3^-$ with mean preformed source signatures for $\delta^{30}Si(OH)_4$ of $1.5 \pm 0.2‰$ (Beucher et al., 2011; Ehlert et al., 2012; Grasse et al., 2013) and for $\delta^{15}NO_3^-$ of $7.1 \pm 0.3‰$ (1SD; Rafter et al., 2012; Rafter and Sigman, 2016) for the EUC. Within the SSCC preformed $\delta^{15}NO_3^-$ values of $5.5 \pm 0.3‰$ (Rafter et al., 2012) are about 1.6‰ lower than the EUC, resulting in an approximate average PCUC value of ~6‰ (Fig. 2a; Mollier-Vogel et al., 2012).

156        The dissolved $\delta^{15}NO_3^-$ of subsurface waters (50-150 m water depth) increases southward (EQ

to 17°S; Mollier-Vogel et al., 2012) as a consequence of water column denitrification, while the dissolved
$\delta^{30}Si(OH)_4$ signature remains close to the source value of 1.5‰ for the PCUC (Fig. 2a; Ehlert et al.,
2012). This difference in the evolution of the isotopic signature from North to South is caused by the
anoxic conditions off Peru only increasing the $\delta^{15}NO_3^-$ signatures via denitrification in the subsurface,
but not affecting the $\delta^{30}Si(OH)_4$ signatures. Accordingly, at the northern shelf between 1°N and 10°S,
where subsurface $O_2$ concentrations $[O_2]$ are >20 μmol $L^{-1}$, N-loss is not observed and the $\delta^{15}N_{bulk}$ values
in the sediments range between 4 and 5‰ close to the $\delta^{15}NO_3^-$ source value of 6‰, thus indicating a high
degree of $NO_3^-$ utilization (Fig. 2b; Mollier-Vogel et al., 2012). In contrast, the $\delta^{30}Si_{BSi}$ signatures north
of 10°S are more variable reflecting an overall lower degree of $Si(OH)_4$ utilization (Doering et al., 2016;
Ehlert et al., 2012). At the central Peruvian shelf (10-12°S), where subsurface $[O_2]$ is <20 μmol $L^{-1}$ (Fig.
2a), the subsurface source value of $\delta^{15}NO_3^-$ increases to 8.6‰ due to denitrification (Mollier-Vogel et
al., 2012). The $\delta^{30}Si_{BSi}$ and $\delta^{15}N_{bulk}$ values both increase as a consequence of higher $Si(OH)_4$ utilization
but a decrease in $NO_3^-$ utilization compared to the northern part of the study area (Fig. 2b) which reflects
the interplay between increased upwelling intensity, high nutrient re-supply and higher consumption via
diatom productivity. In the southernmost part of the shelf (13-16°S), highest productivity and upwelling
intensity prevail today, leading to a further increase in the subsurface $\delta^{15}NO_3^-$ signature of up to 12.5‰
at 15°S, whereas surface sediment mean $\delta^{30}Si_{BSi}$ and $\delta^{15}N_{bulk}$ values further increase reflecting moderate
utilization of both $Si(OH)_4$ and $NO_3^-$ (Figs. 2a-b). The supply of dissolved $Si(OH)_4$ strongly increases
from the northern shelf to the southern shelf area (Fig. 2c), reflecting the strength of the upwelling
conditions. This increase in upwelling and productivity between 10-15°S results in high accumulation
rates of BSi (0.4-0.6 g $cm^{-2}$ $yr^{-1}$; Ehlert et al., 2012) and total nitrogen (TN, 0.026-0.035 g $cm^{-2}$ $yr^{-1}$;
Mollier-Vogel et al., 2012) in the sediment (based on accumulation rates of Gutierréz et al., 2009).
However, the $NO_3^-$ supply, as indicated by subsurface (50-150 m) $NO_3^-$ concentrations in the water
column, slightly decreases from North to South, reflecting the loss of $NO_3^-$ via denitrification.
**2. Sample locations, methods and calculations**
**2.1 Core locations and age models**
The new data in this study were obtained from three short, fine-laminated trigger cores retrieved from
the main upwelling region off the Peruvian margin during the German R/V Meteor cruise M77/2 in 2008
as part of the Collaborative Research Center (SFB) 754 (Fig. 1). New records of $\delta^{30}Si_{BSi}$ and BSi
concentrations were generated for cores M77/2-024-5TC (024-5TC; 11°05'S, 78°00'W, 210 m water
depth), M77/2-005-3TC (005-3TC; 12°05'S, 77°40'W, 214 m water depth) and core M77/2-003-2TC
(003-2TC; 15°06'S, 75°41'W, 271 m water depth). One cm slices of the sediment cores were sampled
for BSi and Si isotope measurements to ensure the availability of sufficient amounts of diatoms for silicon
isotope analysis (Tab. 1). For core 003-2TC additional BSi concentration measurements of material the
extracted from individual laminations was possible (Fleury et al., 2015). As previously published $\delta^{15}N_{bulk}$
values are based on samples from single laminations these were averaged to 1 cm resolution when
directly compared to the $\delta^{30}$Si data in the following. Core locations are shown in Fig. 1. The age models
were published before in Fleury et al. (2015). The age models for all cores are given years AD.

**2.2 Biogenic opal and silicon isotope analyses**

The amount of BSi in the sediments was measured following an automated leaching method using
sodium hydroxide (DeMaster, 1981; Müller and Schneider, 1993) with a precision of 1-2% (1SD).
Unfortunately, no material was left of the cores studied here to estimate dry bulk densities to calculate
mass accumulation rates (MAR). Therefore, MAR values were used from nearby cores BO413 (12 ºS)
and BO406 (14ºS; Gutierréz et al. 2009), which were generally close to 0.02 (g cm$^{-2}$ yr$^{-1}$) during the LIA
and 0.03 g cm$^{-2}$ yr$^{-1}$ during the CWP. The exact bulk MAR values (g cm$^{-2}$ yr$^{-1}$) for each time period were
multiplied by the fractional concentration of BSi and TN (Fleury et al., 2015) to calculate the MAR BSi
and MAR TN (Figs. 2c and 6).
For the Si isotope measurements diatoms were extracted from the sediment by chemical and
physical cleaning (11 and 32 μm sieve; heavy liquid separation with a sodium polytungstate solution set
at 2.15 g mL$^{-1}$) as described in detail in Ehlert et al. (2012; 2013) and Doering et al. (2016). For all
samples the purity of the small diatom fraction (11-32 μm) was evaluated via light microscopy prior to
dissolution and only pure (>95%) diatom samples were treated further. All samples were dissolved in 1
mL 0.1 M NaOH and treated with 200 μL concentrated H$_2$O$_2$ (Suprapur). Sample solutions were diluted
with 4 mL MQ water and neutralized with 0.1 mL 1 M HCl (Reynolds et al., 2008), followed by a
chromatographic purification using 1 mL pre-cleaned AG50W-X8 cation exchange resin (BioRad, mesh
200-400) (de Souza et al., 2012). The Si isotopic compositions were determined in 0.6 ppm sample
solutions on a *NuPlasma HR* MC-ICPMS at GEOMAR applying a standard-sample bracketing method
(Albarède et al., 2004). Silicon isotopic compositions are reported in the δ-notation relative to the
reference standard NBS28 in parts per thousand: $\delta^{30}$Si = ((R$_{sample}$/R$_{standard}$)-1)*1000, where R$_{sample}$ is the
$^{30}$Si/$^{28}$Si ratio of the sample and R$_{standard}$ is the $^{30}$Si/$^{28}$Si ratio of the NBS28. All $\delta^{30}$Si measurements were
run at least in triplicates, with uncertainties ranging between 0.05‰ and 0.27‰ (2SD). Repeated
measurements of an in-house diatom matrix standard gave average $\delta^{30}$Si values of 1.03 ± 0.21‰ (2SD
n=15). Long-term repeated measurements of the reference materials NBS28, IRMM018, and Big Batch
gave average $\delta^{30}$Si values of 0.00 ± 0.24‰ (2SD), -1.40 ± 0.21‰ (2 SD, n=15) and -10.60 ± 0.24‰
(2SD, n=15), respectively, in good agreement with literature values (Reynolds et al., 2007).

**2.3 Diatom assemblage data**

Diatom analysis of cores M77/2-024-5TC, 005-3TC and 003-2TC were published previously based on
three slides per sample and counting of a minimum of 300 valves for each sample (for details see Fleury
et al., 2015). The diatom abundances are presented here for three groups representing different
environmental conditions (Fig. 4e-g): Upwelling species – Chaetoceros sp., Skeletonema costatum,
Thalassionema nitzschioides var. nitzschioides; Coastal planktonic – Actinocyclus spp., Atinoptychus
spp, Asteromphalus spp., and Coscinodiscus sp.; Other diatom species – Nitzschia spp., Rhizosolenia
spp. and Thalassiosira spp., Cyclotella spp., Cocconeis sp.;
The diatom assemblage abundance is compared to $\delta^{30}$Si$_{BSi}$ compositions for cores M77/2-024-
5TC, 005-4TC and 003-2TC to investigate if changes in the assemblage have influenced the isotopic

record. While diatom counts have been performed on bulk sediment samples $\delta^{30}Si_{BSi}$ was measured on the 11-32μm size fraction. However, it was shown previously that this size range closely resembles the main assemblage, which allows studying the influence of changes in the diatom assemblage on the $\delta^{30}Si_{BSi}$ record (Ehlert et al., 2012; 2013).

**2.4 Nutrient utilization**

The degree of nutrient utilization can be described assuming either Rayleigh-type (single input followed by no additional nutrients newly supplied to a particular parcel of water followed by fractional loss as a function of production and export) or steady-state (continuous supply and partial consumption of nutrients causing a dynamic equilibrium of the dissolved nutrient concentration and the product) fractionation behavior (Mariotti et al., 1981). For means of simplification we will only provide the values derived from steady state fractionation, which was shown to better reflect upwelling conditions off Peru (Ehlert et al., 2012).

(1)  $\%Si(OH)_{4\,consumed} = 1 - \left( \left( \delta^{30}Si - \delta^{30}Si(OH)_{4\,source} \right) / ^{30}\varepsilon \right) * 100$

$\%NO_3^-{}_{consumed} = [1 - \left( \delta^{15}N - \delta^{15}NO_3^- \right) / ^{15}\varepsilon ] * 100$

with *%Si(OH)$_4$ consumed* or *%NO$_3^-$ consumed* being the percentages of the supplied Si(OH)$_4$ and NO$_3^-$ that have been utilized. For this calculation we apply enrichment factors of $-1.1‰$ $^{30}\varepsilon$ ($\delta^{30}Si$, (De La Rocha et al., 1997) and $-5‰$ $^{15}\varepsilon$ ($\delta^{15}N$) and assume a constant source water signature of 1.5‰ for $\delta^{30}Si(OH)_4$ source (i.e. the mean $\delta^{30}Si(OH)_4$ of the PCUC). The here calculated nutrient utilization for surface sediments is identical to the original publications (Fig. 2b; Mollier-Vogel et al., 2012; Ehlert et al., 2012). To evaluate the impact of changes in $^{30}\varepsilon$ on the $\delta^{30}Si$ signatures the potential influence of species-specific fractionation was tested based on the impact of a -2.1‰ enrichment factors of *Chaetoceros brevis* (Sutton et al., 2013). However, the estimated impact on past $\delta^{30}Si_{BSi}$ records due to a change in the amount of *Chaetoceros* sp. present in the sediment was less than 5% for all cores (M77/2-024-5TC, 005-3TC and 003-2TC) and thus did not alter the assumed $^{30}\varepsilon$ of -1.1‰ substantially (based on calculations presented in Doering et al., (2016); not shown). The impact of denitrification on the $\delta^{15}NO_3^-$ signatures of the past is assessed in the following section before calculating past NO$_3^-$ utilization for the respective latitudes.

**2.5 Calculation of the $\delta^{15}NO_3^-$ source signatures**

Based on modern observations from the water column it is known that NO$_3^-$ and Si(OH)$_4$ are incorporated in a 1:1 ratio when diatoms dominate the phytoplankton assemblage (Brzezinski, 1985; Ragueneau et al., 2000). The ratio of nutrients in the water column can, however, vary between 2:1 and 1:2 on the shallow Peruvian shelf (Grasse et al., 2016). Assuming a strict 1:1 uptake ratio of nutrients, the respective $\delta^{30}Si_{BSi}$ and $\delta^{15}N_{Bulk}$ values in the underlying sediments should also reflect a 1:1 ratio (indicated by white star '1' in Fig. 3a). Based on the known $\delta^{30}Si(OH)_4$ and NO$_3^-$ source signatures of modern subsurface waters, we can calculate the actual nutrient utilization (see section 2.4) and estimate the uptake ratio for NO$_3^-$ :Si(OH)$_4$ (Fig. 3b). However, it is not possible to observe a significant correlation for the entire shelf area, given that there are only a few data points for the areas along the shelf (Figs. 2, 5a). We calculate past nutrient utilization and estimate the influence of denitrification on the $\delta^{15}N_{bulk}$ values based on the fact that on the shelf $\delta^{30}Si_{BSi}$ and $\delta^{15}N_{bulk}$ values generally follow a positive linear regression (Figs. 3b,

5a). In order to estimate past changes in the $\delta^{15}NO_3^-$ source values, the $\delta^{30}Si_{BSi}$ and $\delta^{15}N_{bulk}$ values were separately plotted against each other for the time periods of the CWP and the LIA, arid and humid (Fig. 5 b-d). Accordingly, the $\delta^{15}NO_3^-$ source value for each period was calculated based on the linear function assuming that the source $\delta^{30}Si(OH)_4$ signature always remained stable at 1.5‰ over time:

(2) $\delta^{30}Si(OH)_4 = a * \delta^{15}NO_3^- + b$, or

(3) $\delta^{15}NO_3^- = \left(\delta^{30}Si(OH)_4 - b\right)/a$

with *a* indicating the slope of the line and b the intercept. For $\delta^{30}Si(OH)_4$ we used the value of 0.4‰ representing near-0% utilization (= source water $\delta^{30}Si(OH)_4$ of 1.5‰ of the PCUC – 1.1‰ fractionation during uptake) to estimate the $\delta^{15}NO_3^-$ source. Accordingly, the values estimated by equation (3) represent the $\delta^{15}NO_3^-$ source value assuming also near-0% utilization of $NO_3^-$

We calculated the linear regression based on all samples of the different cores from the different latitudes (11°S, 12° S, 14°S and 15°S) during the CWP and LIA. We also further differentiated between short-term productive phases (arid phases) and the generally prevailing humid El-Niño like conditions during the LIA (grey shadings in Fig. 4), and resolved the resulting equation based on eq. (2) to estimate $\delta^{15}NO_3^-$. Only for the LIA (humid) phases it was not possible to directly calculate $\delta^{15}NO_3^-$ values based on the linear function from eq. (2), due to near horizontal alignment of the $\delta^{15}N_{bulk}$ versus $\delta^{30}Si_{BSi}$ values (Fig. 5b). Therefore, for this time period the highest $\delta^{15}N_{Bulk}$ value for each latitude was assumed to reflect ~100% utilization and was thus used as $\delta^{15}NO_3^-$ source value. This assumption might slightly overestimate the maximum utilization, which is only ~80% today (Fig. 2b), and therefore might underestimate the source value slightly. For all time periods and latitudes, the linear regressions as well as correlation coefficient ($r^2$) are given in the supplements (Fig. S1). The results are presented in the following as the resulting $\delta^{15}NO_3^-$ source values and the theoretical ratio of nutrient utilization (i.e. 1:1 or 2:1, 15:1, etc.; Fig. 5. b-d) for each latitudinal range to compare the latitudinal trends between the CWP and LIA.

**2.6 Calculation of Nutrient supply**

Based on these calculated subsurface $\delta^{15}NO_3^-$ values we further calculated the change in nutrient utilization as well as nutrient supply for the different latitudes. Past nutrient utilization was calculated following equation (1). Given the estimate of nutrient demand and export productivity it is further possible to estimate changes in the supply with the relationship Δsupply = Δdemand/Δutilization by applying the equation of (Horn et al., 2011) given by:

(4) $Nutrient\ supply = \dfrac{F_{BSi/TN}^{sample}/F_{BSi/TN}^{present}}{\%nutrient_{consumed}^{sample}/\%nutrient_{consumed}^{present}}$

$F_{opal/TN}$ is the flux of BSi or TN and *%nutrient$_{consumed}$* is the percent of the $Si(OH)_4$ or $NO_3^-$ supply consumed (i.e. nutrient utilization). Given that there are no accumulation rates available for either the surface sediment samples or for any of the cores studied here to directly determine the export productivity directly, we used the BSi and TN values previously published (surface sediments; Mollier-Vogel et al., 2012; Ehlert et al., 2012) and the new BSi values presented in this study together with mass accumulation rates (g cm$^{-2}$ yr$^{-1}$) for cores BO406-13 and 406-5 from Gutierréz et al. 2009 to calculate the accumulation rates of BSi and TN (MAR; g cm$^{-2}$ yr$^{-1}$). For the different time periods mean values for MAR BSi and

MAR TN were calculated and the respective nutrient supply was calculated based on equation (4),
indicating changes in the nutrient supply compared to modern values.
**3. Results**
**3.1 Biogenic opal and silicon isotope signatures**
The data of the sediment cores from the shelf area between 12°S and 15°S presented here show an
increase in BSi content from mean values of 13-23% during the LIA to values of 21-29% during the
CWP. The $\delta^{30}$Si records follow a similar trend of lower mean $\delta^{30}$Si$_{BSi}$ values of 0.8 ± 0.2‰ (2SD, 12°S),
0.8 ± 0.1‰ (14°S) and 1 ± 0.2‰ (15°S) during the LIA to more variable and higher mean values of 1.3
± 0.4‰ (12°S), 0.8 ± 0.4‰ (14°S) and 1.5 ± 0.2‰ (15°S) during the CWP (Fig. 3 a-d; Table 1).

317          The diatom assemblages (Fig. 4e-g; based on Fleury et al., 2015) show an association of the

amount of upwelling species and $\delta^{30}$Si$_{BSi}$ signatures, with decreases of up to 20% in upwelling species
often accompanied by a reduction of $\delta^{30}$Si$_{BSi}$ by about 0.5-1‰. However, not every decrease in $\delta^{30}$Si$_{BSi}$
is mirrored by a change in the diatom assemblage and vice versa (e.g. Fig. 4f at 1650 AD). Overall the
diatom assemblage data indicate little changes in the mean conditions and a slight reduction of upwelling
strength at 12°S and 15°S during the LIA in comparison to the CWP (Fig. 6). The most distinct shift of
lower abundances of upwelling species (~50%) to higher values during the CWP (~70%) is found at
15°S (003-2TC) corresponding to the strongest changes in BSi and $\delta^{30}$Si$_{BSi}$ at this location.

325          The sedimentary BSi concentrations and $\delta^{30}$Si$_{BSi}$ signatures at 12°S (005-3TC) and 15°S (003-

2TC) were lowest during the LIA (Fig. 4c,e), in agreement with previously published records from 11°S
(M77/1-470; Fig. 4a) and 14°S (Ehlert et al., 2015; Fig. 4d). An exception is core 024-5TC (Fig. 4a) from
11°S, where $\delta^{30}$Si mean values of the LIA (1.3 ± 0.4‰) are similar to CWP mean values (1.4 ± 0.1‰).
Furthermore, both the BSi concentrations and $\delta^{30}$Si$_{BSi}$ signatures of core 024-5TC were significantly
higher during the LIA than at nearby core M77/1-470 (Fig. 4a; Ehlert et al. 2015). However, comparison
with the cumulative diatom assemblage indicates overall little difference in the amount of upwelling and
coastal planktonic diatom species between the LIA and the CWP at 11°S (Fig. 4f), with intervals of
reduced abundances of upwelling species generally lasting less than 50 years, much shorter than the 100
to 150 years intervals observed at 12°S and 15°S. Furthermore, the finely laminated sediment layers do
indicate short periods of higher productivity during the LIA in phase with more arid conditions (Fig. 4,
grey shadings; for details see Fleury et al., 2015). Accordingly, the high mean BSi and $\delta^{30}$Si$_{BSi}$ values
obtained from core 024-5TC may be an artifact of low sampling resolution with only two $\delta^{30}$Si$_{BSi}$ samples
representing the time period between 1700 and 1800 AD and $\delta^{30}$Si$_{BSi}$ analyses not evenly covering all
the short events (~50 years) of reductions in the abundance of upwelling diatom species (Fig. 4f).
Alternatively, the increase in Si(OH)$_4$ utilization decoupled from an increase in diatom abundance
(Fleury et al., 2015; not shown here) may indicate stronger silicification of the diatom frustules, as often
observed under Fe-deficient conditions and associated with an increase in the Si(OH)$_4$:NO$_3^-$ incorporated
by the diatoms (De La Rocha et al., 2000; Takeda, 1998; Wilken et al., 2011).

344          As previously shown the $\delta^{15}$N$_{bulk}$ values of the three cores (M77/2-024-5TC, 005-3TC and 003-

2TC) presented in this study were on average 0.8‰ lower during the LIA than during the CWP (Fleury
et al., 2015). The $\delta^{15}N_{bulk}$ values reported for core 005-3TC (12°S) are close to values of nearby core
B0406-13 (Gutiérrez et al., 2009). Similarly, the $\delta^{15}N_{bulk}$ values of core 003-2TC (15°S) agree well with
previously published $\delta^{15}N_{bulk}$ record of core B0405-6 (14°S, Fig. 4 j, k; Gutiérrez et al., 2009).

### 3.2 $\delta^{15}NO_3^-$ source signatures, nutrient utilization and supply

During the humid phases of the LIA the calculated $\delta^{15}NO_3^-$ source values were lower reaching values of
6‰ between 11°S and 12°S and 7.5‰ between 14°S and 15°S (Fig. 5b, 6a). The calculated $NO_3^-$
utilization was higher during this time reaching values between 70 and 90%, while $Si(OH)_4$ utilization
ranged between 6 and 60%. The MAR TN was lowest (<0.02 g cm$^{-2}$ yr$^{-1}$) during the LIA, however, with
little difference between humid and arid phases (Fig. 6 a+c, right side). The MAR BSi values were similar
to today during the LIA (humid) ranging between 0.2 - 0.5 g cm$^{-2}$ yr$^{-1}$ (Figs. 2, 6a right side). The
calculated $NO_3^-$ supply was lowest during the LIA (humid) ranging between 0.3 and 0.7 with little change
over latitude in accordance with the prevalence of more oxygenated waters, whereas the $Si(OH)_4$ supply
strongly increased from 0.5 to 3.8 at 12°S (Fig. 6a).
During the CWP the calculated $\delta^{15}NO_3^-$ source signatures based on eq. (2) and (3) result in
values of 7.6‰ at 11°S, 8.6‰ at 12°S and 10.4‰ between 14°S and 15°S during the CWP (Figs. 5c, 6b;
S1), which reflects a southward increase in $\delta^{15}NO_3^-$ source signatures as observed today (Fig. 2a). Based
on these $\delta^{15}NO_3^-$ values the nutrient utilizations estimated based on eq. (1) range between 30-90% for
$NO_3^-$ and 40-100% for $Si(OH)_4$ (Fig. 6b). During the LIA (arid) similar values are calculated with a
$\delta^{15}NO_3^-$ mean value of 8‰ between 11°S and 12°S, increasing to a value of 9‰ between 14°S and 15°S
(Fig. 6c). The respective nutrient utilization ranges between 2 to 70% for $NO_3^-$ and 20 to 85% for
$Si(OH)_4$. The MAR TN were by about 0.02 (g cm$^{-2}$ yr$^{-1}$) lower during the CWP than today and MAR BSi
values were generally higher by about 0.1 - 0.35 (g cm$^{-2}$ yr$^{-1}$). The calculated $Si(OH)_4$ supply indicates a
slight increase compared to today (as indicated by positive values) but has remained rather stable around
1 over all latitudes, while the and $NO_3^-$ supply also indicate values of ~1 at 11°S and 15°S but the supply
increased to 2 at 12°S (Fig. 6b, right side) . During the LIA (arid) MARs of TN and BSi were both lower
in comparison to the CWP, ranging between 0.014 - 0.017 (g cm$^{-2}$ yr$^{-1}$) and 0.25 - 0.7 (g cm$^{-2}$ yr$^{-1}$),
respectively. The $Si(OH)_4$ supply was similarly stable as observed during the CWP but slightly higher
ranging from 0.6 to 1.55, while the $NO_3^-$ supply was lower and decreased from North to South from 1 to

374  0.3.

### 4. Discussion

The aim of this study is to reconstruct the extent of variability in $\delta^{15}N_{bulk}$ caused as a function of
denitrification versus nutrient utilization during specific time periods, i.e. the CWP and recurring short-
term arid/humid periods during the LIA. The combination of $\delta^{15}N_{Bulk}$ and $\delta^{30}Si_{BSi}$ signatures enables us
to calculate the $\delta^{15}NO_3^-$ source signatures during these time periods of time, enabled us to estimate the
extent of $NO_3^-$ utilization additional contributing to the $\delta^{15}N_{bulk}$ recorded in the sediments. These data are
combined with the $\delta^{30}Si_{BSi}$ signatures, calculated $Si(OH)_4$ utilization calculations and nutrient supply,
will be discussed in the following (1) in comparison to modern conditions, and (2) in the context of
consistency with ENSO variability observed off Peru and the Eastern Equatorial Pacific (EEP) during
the last 600 years. Due to similar conditions prevailing during the CWP and arid phases of the LIA we
will discuss these time periods together in the following.

**4.1. Disentangling nutrient supply, utilization and N-loss processes: Changes in the source water**
**nitrate isotopic composition**

**Humid conditions of the Little Ice Age**

During the humid LIA the $\delta^{30}Si_{BSi}$ values remain remarkably stable, whereas $\delta^{15}N_{bulk}$ values show a wide
range potentially reflecting enhanced $NO_3^-$ limitation prevailing during humid phases (Fig. 5b). Such a
shift towards increasing $\delta^{15}N_{bulk}$ values with consistently low $\delta^{30}Si_{BSi}$ values is indicative of weaker
denitrification due to the higher subsurface oxygenation (only suboxic and not anoxic conditions) in
agreement with reconstructions of redox conditions (Salvatteci et al., 2014b; Sifeddine et al., 2008). This
in in agreement with the lower $\delta^{15}NO_3^-$ source signatures (6-7.5 ‰) and a decrease in the abundance of
upwelling-indicating diatom species and *Chaetoceros* sp. (Figs. 4e-f and 6a; data from Fleury et al.,
2015). Furthermore, our results indicate much higher $NO_3^-$ utilization over $Si(OH)_4$ utilization with ratios
of up to 15:1 (Fig. 5b, 7a). This is in agreement with phytoplankton assemblage analyses during El-Niño
events when productivity has been reported to be dominated by non-siliceous phytoplankton groups
(Sanchez et al., 2000), which is also observed today further off the coast of Peru (Fig. 3a; Grasse et al.,
2016). Accordingly, with prevalence of non-siliceous phytoplankton groups more $NO_3^-$ than $Si(OH)_4$ is
utilized (Conley and Malone, 1992; Wilkerson and Dugdale, 1996) and the ratio might shift to ratios of
up to 15:1 (Fig. 3b, 7a; Grasse et al., 2016). However, the conditions found offshore today are based on
surface waters that originate from the shelf area where diatom blooms prevail, thus being already
depleted in $Si(OH)_4$ and might not provide an adequate analogue for the conditions prevailing during the
humid LIA phases (Fig. 5b, 6a). The calculated $NO_3^-$ supply was lowest with little change over latitude
in accordance with prevalence of more oxygenated waters, whereas the $Si(OH)_4$ supply strongly
increased especially at 12°S (Fig. 6a). However, the calculated increased $Si(OH)_4$ supply likely reflects
the change in nutrient uptake (i.e. nutrient ratio) due to stratification and potentially Fe limitation rather
than an actual increase in $Si(OH)_4$ supply reaching surface waters. Accordingly, we observe a high
$Si(OH)_4$ supply but low utilization, reflecting a low $Si(OH)_4$ demand at the time. The $NO_3^-$ supply appears
to be lower than today but the strongly enhanced $NO_3^-$ utilization indicates a higher $NO_3^-$ demand. This
shift towards a decreased $Si(OH)_4$ but an increased $NO_3^-$ demand further supports a change in the nutrient
uptake ratio by phytoplankton ($NO_3^-$: $Si(OH)_4$ = 2:1 or 15:1, Fig. 5b, 7a). Regarding the high Si supply
it is also possible that is was actually bound by non-siliceous phytoplankton species, such as
Synechococcus and not by diatoms as observed further offshore today (Fig 3a; Grasse et al., 2016).
However, these species are more likely to be recycled within the water column and Si stored within their
cells is thus remineralized and not transported to the sediment. This might be the reason we observe low
BSi (%) values and the $\delta^{30}Si_{BSi}$ remain equally low (Fig. 4, 5b).

**Current Warm Period and arid phases of the Little Ice Age**

The calculated $\delta^{15}NO_3^-$ source values based on linear regression between $\delta^{15}N_{Bulk}$ and $\delta^{30}Si_{BSi}$ indicate
an increase in the $\delta^{15}NO_3^-$ source signatures of upwelled subsurface waters from North to South from

~7‰ to 10‰ during the CWP and arid phases of the LIA similar to those observations for modern conditions (Figs. 2a, 5c-d, 7b). This is in agreement with high contributions of upwelling diatoms and *Chaetoceros* sp. during both time periods (Figs. 4e-g. and 6b-c; data from Fleury et al., 2015). The calculated $Si(OH)_4$ and $NO_3^-$ supplies indicate a slight increase compared to today with $Si(OH)_4$ supply increasing and $NO_3^-$ supply decreasing towards the southern shelf. The latter agrees with continuous denitrification in the southern area causing loss of $NO_3^-$. Furthermore, nutrient utilization for both $NO_3^-$ and $Si(OH)_4$ were moderate to high (~30-90%; Fig. 6b-c, 7b), similar to modern values between 10°S and 15°S (Fig. 2b; Mollier-Vogel et al., 2012; Ehlert et al., 2012). Due to the incomplete utilization of $NO_3^-$ the increasing $\delta^{15}NO_3^-$ source values are also only partially reflected in the $\delta^{15}N_{Bulk}$ signatures for the CWP and LIA (arid) as previously reported for signatures from surface sediments (Mollier-Vogel et al., 2012). Especially during the CWP, we calculate about 20-40 % lower $NO_3^-$ utilization compared to today (Fig. 2b) but at the same time $NO_3^-$ supply increased, while $Si(OH)_4$ supply was only slightly higher compared to today and remained rather stable with latitude (Fig. 6b, right side). Apparently, the nutrient concentration of upwelled waters during the CWP has been different from today, which is also supported by a difference in the ratio of $NO_3^-$:$Si(OH)_4$ utilization (Fig. 5c). Accordingly, unlike todays surface sediment data, the cores at 11°S and 12°S show substantially higher $\delta^{30}Si_{BSi}$ values during both the CWP and LIA (arid) (Fig. 5c-d). These higher $\delta^{30}Si_{BSi}$ signatures result in a $NO_3^-$:$Si(OH)_4$ utilization that has shifted towards a 1:2 ratio, indicating enhanced utilization of $Si(OH)_4$ over $NO_3^-$ potentially leading to $Si(OH)_4$ limitation, in agreement with the lower $Si(OH)_4$ supply in comparison to $NO_3^-$ and higher $Si(OH)_4$ than $NO_3^-$ utilization rates (Fig. 6b-c). Such a decoupling of Si and N within diatoms can be caused by biogeochemical changes, such as Fe availability altering the Si:N uptake dynamics (Hutchins and Bruland, 1998; Takeda, 1998) whereby elevated Si:N ratios are characteristic for Fe-limited diatom communities (Takeda, 1998). Accordingly, increased uptake of Si over N can lead to a $Si(OH)_4$ limitation as found during the CWP and LIA arid phases at 11°S to 12°S (Figs. 5c-d, 6b-c, 7b). The reason may have been that less Fe was upwelled at the narrow shelf between 11°S and 16°S, which led to Fe-limitation during progressing diatom blooms (Doering et al., 2016).

**4.2 The coupling between the biogeochemical cycle and ENSO variability**

Recent evidence shows that a cool EEP plays a key role in climate change due to its linkage to a slowdown in global warming (England et al., 2014; Kosaka and Xie, 2013) highlighting the importance to understand Pacific climate variability in the past (Rustic et al., 2015). The last millennium has been divided into warmer global conditions over the Medieval Warm Period (MWP), colder temperatures over the LIA and rising temperatures since the beginning of the CWP (Mann et al., 1999). The transition between the MWP and LIA (~1150 to 1500 AD) has been associated with an anomalous strong zonal SST gradient and with transitional Northern Hemisphere (NH) cooling into the LIA as evidenced by cooler SSTs at Galápagos (Rustic et al., 2015). After ~1500 AD, the EEP cooling trend ended and local SSTs began to increase until around 1600 AD an anomalous weak zonal gradient was established when the EEP temperatures reversed from cooling to warming. This reversal occurred when the NH descended into the coldest part of the LIA and persisted throughout most of the LIA resulting in an extended El Niño-like mean state (Mann et al., 2009). Evidence links the ITCZ to hemispheric warming and cooling cycles implying southward ITCZ displacements during NH cold periods (Chiang and Bitz, 2005;

Schneider et al., 2014). Accordingly a southward shift of the ITCZ during the MWP-LIA transition has been proposed for the Atlantic and Pacific (Haug et al., 2001; Peterson and Haug, 2006; Sachs et al., 2009). The El Niño-like conditions during the LIA have been associated to a gradual intensification of the fluvial input of sediments to the continental shelf as reflected by an increase in the terrigenous sediment flux (Briceño-Zuluaga et al., 2016; Gutiérrez et al., 2009; Sifeddine et al., 2008), changes of the radiogenic isotopic composition of the terrigenous fraction due to changes in the provenance and material transport (Ehlert et al., 2015), as well as better oxygenation and a lower productivity in the Peruvian upwelling area (Gutiérrez et al., 2009; Salvatteci et al., 2014b; Sifeddine et al., 2008). Accordingly, most of the LIA i.e., the humid phases, have been characterized by low productivity and weak denitrification intensity between 10°S and 15°S (Díaz-Ochoa et al., 2009; Salvatteci et al., 2014b; Sifeddine et al., 2008), which is supported by the absence of a significant southward increase in the source value of $\delta^{15}NO_3^-$ reconstructed from our records (Fig. 5b, 6a). Correspondingly, high $\delta^{15}N_{bulk}$ and little change in reconstructed $NO_3^-$ supply indicate more complete $NO_3^-$ utilization during the LIA (humid), while $\delta^{30}Si_{BSi}$ signatures and utilization remained low and Si supply high (Fig. 6a). This indicates a shift towards a dominance of non-siliceous phytoplankton productivity causing $NO_3^-$ limitation and low uptake of Si. This is in agreement with modern conditions during El Nino events for which physical-biogeochemical models together with in-situ and satellite observations (1958-2008) have shown that the temperatures and sea level increase, the thermocline/nutricline deepens, and the phytoplankton (mainly diatoms) and nutrient concentration decrease along the Peruvian coast (Espinoza-Morriberón et al., 2017). Coastal trapped waves propagating along the coast can seasonally increase the depth of thermocline and nutricline, decreasing the $NO_3^-$ vertical flux into the surface layer. The $NO_3^-$ and Fe content of the upwelling source waters may also strongly decline (Espinoza-Morriberón et al., 2017). Our calculations show that phases of lower productivity during so-called El Niño like-conditions during the LIA (humid), indeed have a $\delta^{15}NO_3^-$ source delivered to the Peru upwelling area similar to today (Fig. 6a), but due to less or no denitrification $\delta^{15}NO_3^-$ source does not increase southward (Fig. 5b 6a). Instead $\delta^{15}N_{Bulk}$ values are mainly affected by variability of $NO_3^-$ concentrations, which seem to be the limiting factor for PP similar as observed during El Niño events today (Espinoza-Morriberón et al., 2017). Potentially the stronger stratification due to deepening of the nutricline does not allow for similarly efficient N remineralization (or N is transported offshore due to eddy activity, Espinoza-Morriberón et al., 2017) and may result in $NO_3^-$ to be utilized more strongly than $Si(OH)_4$ (Fig. 7a). Accordingly, the $NO_3^-$ supply was diminished while $Si(OH)_4$ was still available.

Finely laminated sediment from the LIA from the Oxygen Minimum Zones (OMZ), which have also been used in this study, resolve multidecadal variations in precipitation over the continent, and of variations in detrital and biogenic fluxes in relation to precipitation and upwelling intensity (Briceño-Zuluaga et al., 2016, Díaz-Ochoa et al., 2009; Fleury et al., 2015; 2016; Salvatteci et al., 2014a). And stable oxygen isotope compositions of individual planktic foraminifera point to greater ENSO activity (high frequency between alternating La Niña and El Niño conditions) in the EEP based on records from Galapagos (Rustic et al., 2015). It was shown by coupled models that such multi-decadal variation in ENSO amplitude can arise from episodic strengthening and weakening of the thermocline feedback (Borlace et al., 2013). The difference we observed in the isotopic evolution of nutrients ($\delta^{15}N$ and $\delta^{30}Si$) between the arid and humid phases during the LIA support the development of multidecadal phases of

prevailing La Niña- or El Niño-like conditions. Similar interannual variance has been observed based on organic carbon and carbonate proxies during the LIA within the California current system, which the authors related to large ENSO events in contrast to an apparent reduction in such variability during the CWP (Abella-Gutiérrez and Herguera, 2016).

This El Niño-like mean state appears to have ended at the beginning of the CWP (Rustic et al., 2015). Evidence for increasing precipitation off the coast of Panama after 1700 AD likely reflects the northward shift of the ITCZ (Linsley et al., 1994) from its more southerly LIA position. During the CWP the OMZ intensified and marine productivity increased together with surface temperature cooling and increase in terrigenous material input (Briceño-Zuluaga et al., 2016; Gutiérrez et al., 2011). We find that the CWP and LIA (arid), are characterized by high upwelling intensity, productivity and N-loss processes (Fleury et al., 2015; Salvatteci et al., 2014b; Sifeddine et al., 2008), are associated with southward increasing $\delta^{15}NO_3^-$ source signatures caused by denitrification, reflecting moderate $NO_3^-$ utilization and moderate to high $Si(OH)_4$ utilization (Fig. 7b). Highest $\delta^{30}Si_{BSi}$ and utilization values at 15°S are potentially caused by progressive Fe limitation during diatom blooms, causing a $NO_3^-$:$Si(OH)_4$ ratio of up to 1:2. Southward increasing $\delta^{15}N_{bulk}$ values and calculated $\delta^{15}NO_3^-$ demonstrate the consistent incorporation of higher isotopic compositions due to subsurface denitrification under anoxic subsurface conditions in agreement with decreasing $NO_3^-$ supply illustrating the N-loss process.

**Conclusions**

Based on a compilation of new and previously published $\delta^{30}Si_{BSi}$ and $\delta^{15}N_{bulk}$ records of several short sediment cores from the southern Peruvian shelf (11-15°S) we present a new evaluation of the impact of denitrification on the isotopic source signature of $NO_3^-$ and subsequent utilization. As denitrification increases southward along the shelf today, we applied a latitudinal comparison between $\delta^{30}Si_{BSi}$ and $\delta^{15}N_{bulk}$ signatures in modern surface and latest Holocene sediments. Given that during the last 600 years both proxies have mainly been influenced by nutrient utilization we performed a novel calculation of subsurface $\delta^{15}NO_3^-$ based on the linear regression of $\delta^{30}Si_{BSi}$ and $\delta^{15}N_{bulk}$ signatures for the CWP and LIA (arid versus humid conditions). Our results show that low productivity and higher subsurface oxygenation (suboxic conditions) during the humid phases of LIA were associated with low $\delta^{30}Si_{BSi}$ and $\delta^{15}N_{bulk}$ signatures. The latitudinal comparison of $\delta^{30}Si_{BSi}$ versus $\delta^{15}N_{bulk}$ signatures supports decreased influence of subsurface denitrification on the $NO_3^-$ isotope distribution with lower and more uniform $\delta^{15}NO_3^-$ source signatures between 6 and 7.5‰. However, $NO_3^-$ utilization was significantly higher, while $Si(OH)_4$ utilization was lower because the $Si(OH)_4$ supply was higher compared to the demand. This change in nutrient utilization is reflected by a $NO_3^-$:$Si(OH)_4$ uptake ratio of up to 15:1, suggesting a shift from a diatom-dominated regime to one dominated by non-siliceous phytoplankton. This agrees with El-Niño-like conditions prevailing during most of the LIA accompanied by a deepening of the thermocline and lower nutrient availability. During the CWP and sporadic arid conditions during the LIA the isotopic compositions of $NO_3^-$ have increased southward due to subsurface denitrification under strong oxygen depletion, similar to modern conditions. Furthermore, enhanced $Si(OH)_4$ over $NO_3^-$ uptake characterized nutrient utilization over $NO_3^-$ uptake, reflecting strong diatom blooms as observed today, potentially leading to progressive Fe limitation increasing the Si:N uptake ratio of diatoms to 2:1.

In summary, our results constitute an improvement of the application of combined $\delta^{30}Si_{BSi}$ and
$\delta^{15}N_{bulk}$ signatures as a powerful tool to differentiate between past changes in subsurface denitrification,
nutrient utilization and supply but also changes in the nutrient ratios as a result of either micro-(Fe) or
macro-nutrient limitation.
**Data availability**
All data will be uploaded at www.pangea.de upon publication
**Author contributions**
S. Fleury and K. Doering conducted the sampling of the sediment cores at Bordeaux University. K.
Doering prepared the samples and performed the isotope measurements. K. Doering wrote the
manuscript with contributions from all co-authors.
**Acknowledgement**
This work is a contribution of the Collaborative Research Centre 754 "Climate-Biogeochemistry
interactions in the Tropical Ocean" (www.sfb754.de), which is supported by the Deutsche
Forschungsgemeinschaft (DFG).

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

**Table 1: Downcore record of core M77/2-024-5TC, M77/2-005-3TC and M77/2-003-2TC for $\delta^{30}Si_{BSi}$ (‰) and**
**BSi content (wt%). The 2 SD represents the external reproducibility of repeated sample measurements.**

| Core | Age yrs BP | Depth (mm) | BSi (wt%) | $\delta^{30}Si_{BSi}$ (‰) | 2SD |
|---|---|---|---|---|---|
| 24-5TC | 42 | 0 | 16.2 | 1.50 | 0.23 |
| | 101 | 42 | 16.1 | 1.26 | 0.17 |
| | 154 | 104 | 34.3 | 1.50 | 0.18 |
| | 170 | 134 | 29.3 | 1.43 | 0.15 |
| | 187 | 161 | 23.7 | 1.47 | 0.05 |
| | 243 | 213 | 30.7 | 1.35 | 0.21 |
| | 304 | 264 | 28.1 | 1.40 | 0.09 |
| | 376 | 301 | 21.0 | 1.38 | 0.16 |
| | 422 | 390 | 10.1 | 0.81 | 0.19 |
| | 441 | 432 | 24.6 | 1.51 | 0.16 |
| | 483 | 473 | 23.8 | 1.61 | 0.08 |
| 005-3TC | 46 | 0 | 15.9 | 1.07 | 0.09 |
| | 73 | 35 | 15.0 | 1.37 | 0.11 |
| | 95 | 69 | 25.4 | 1.46 | 0.21 |
| | 217 | 128 | 18.8 | 1.03 | 0.18 |
| | 250 | 165 | 17.3 | 0.80 | 0.22 |
| | 259 | 185 | 15.1 | 0.93 | 0.13 |
| | 303 | 241 | 13.1 | 0.44 | 0.27 |
| | 340 | 296 | 14.0 | 0.50 | 0.15 |
| | 358 | 323 | 11.6 | 0.47 | 0.20 |
| | 450 | 369 | 14.5 | 1.24 | 0.24 |
| | 464 | 389 | 25.0 | 1.60 | 0.19 |
| 003-2TC | 22 | 0 | 39.2 | 1.63 | 0.24 |
| | 146 | 97 | 40.5 | 1.48 | 0.05 |
| | 245 | 174 | 41.9 | 1.30 | 0.26 |
| | 288 | 208 | 20.8 | 0.65 | 0.23 |
| | 327 | 239 | 23.9 | 0.74 | 0.13 |
| | 411 | 304 | 19.4 | 0.73 | 0.27 |
| | 474 | 353 | 46.7 | 1.38 | 0.17 |
| | 581 | 437 | 29.1 | 0.63 | 0.12 |



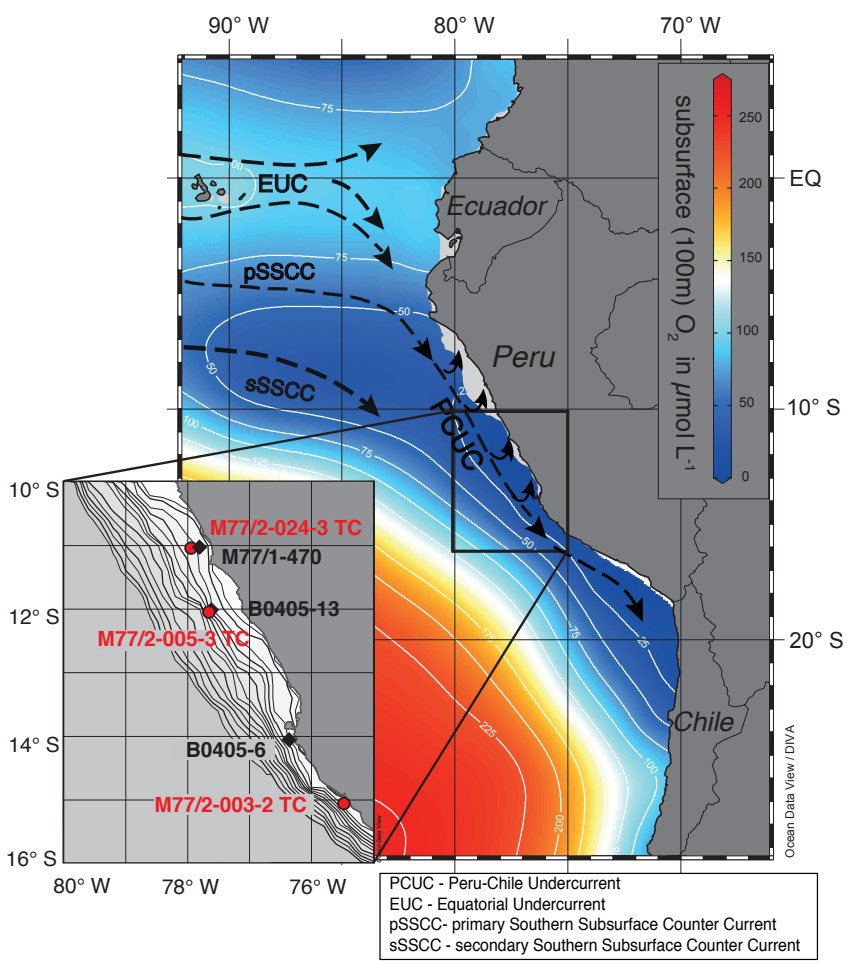


**Figure 1: Subsurface (100 m) oxygen concentration and current directions in the Eastern Equatorial Pacific.**
**Inset map shows locations of cores M77/2-024-3 TC, M77/2-005-3 TC, M77/2-003-2 TC (this study) and**
**M77/1-470, B0405-13 and B0405-6 (Ehlert et al., 2015; Gutiérrez et al., 2009) in more detail. The bathymetry**
**is given for 0 to 1000 m water depth in 50 m increments.**

812

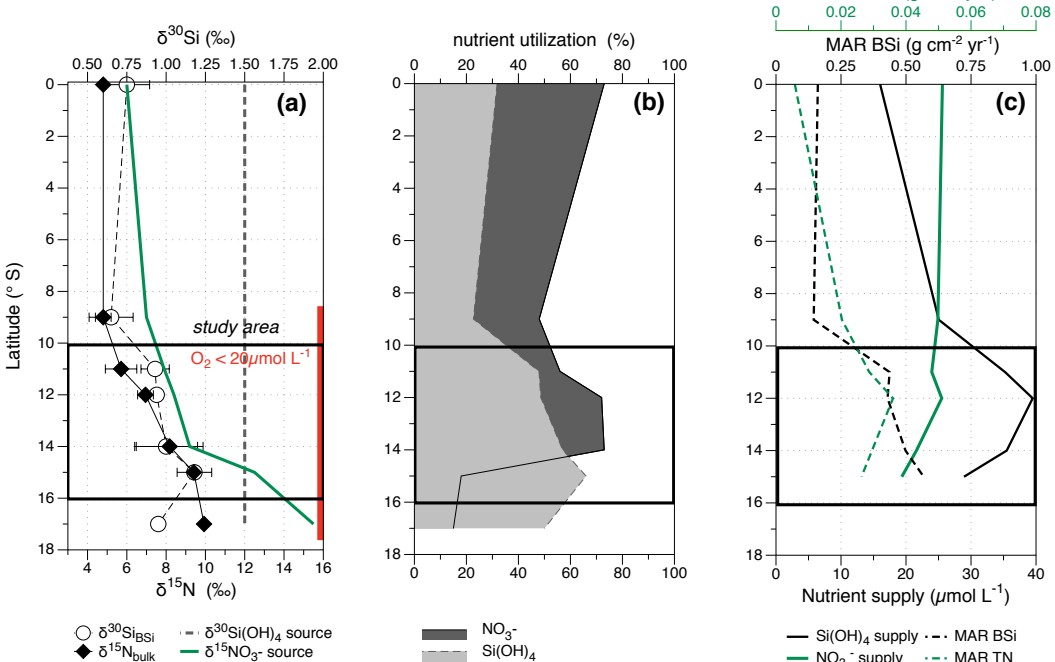

813

**Figure 2: Latitudinal overview of present day (a) mean δ15N_bulk (‰, black diamonds, 2 SD error bars) and**
**δ30Si_BSi (‰, white circles, 2SD error bars), the black dashed line indicates the subsurface δ30Si(OH)4 source**
**value of 1.5 ‰, the green solid line marks the δ15NO3- source value, increasing southwards from 6‰ (EQ-8°S),**
**to about 8‰ (10-12°S) and 12.5‰ (15°S). The red bar indicates the area of suboxic conditions in subsurface**
**waters. The black rectangle marks the study area for downcore reconstruction (see also Fig.1). (b) Nutrient**
**utilization for NO3- (%, dark grey area) and Si(OH)4 (%, dashed area). (c) MAR TN (g cm-2 yr-1) and MAR**
**BSi (g cm-2 yr-1; for calculation see section 2.6) and nutrient supply (modified after Mollier-Vogel et al., 2012**
**and Ehlert et al., 2012).**









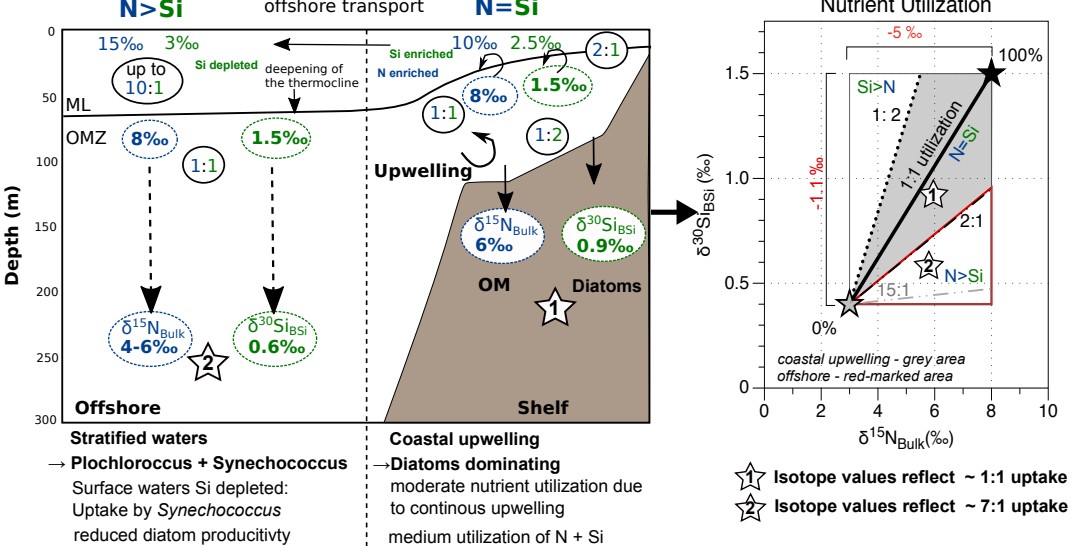


**Figure 3: a. Simplified schematic figure of the 10°S transect off Peru indicating the concentrations of Si(OH)₄**
**(green) and NO₃- (blue; given as enriched or depleted) together with the NO₃⁻:Si(OH)₄ ratios (N:Si ). The**
**stable isotope composition in the water column given as δ³⁰Si(OH)₄ and δ¹⁵NO₃⁻ and the δ³⁰Si_BSi and δ¹⁵N_Bulk**
**signatures in underlying sediments. Diatoms are dominant on the shelf, whereas non-siliceous organisms**
**(Synechococcus, Prochlorococcus) dominate the offshore productivity (modified from Grasse et al., 2016);**
**b. Schematic overview of nutrient utilization, the black star marks the source signature (or 100% utilization)**
**for δ¹⁵NO₃⁻ (8‰) and for δ³⁰Si(OH)₄ (1.5‰) at the location, the grey star marks the theoretical isotopic**
**compositions for ~0% utilization, the thick black solid line indicates the 1:1 utilization for N=Si, respectively.**
**The δ³⁰Si_BSi and δ¹⁵N_Bulk signatures from the shelf (white star 1) reflect an N:S uptake close to 1:1 while**
**offshore signatures (white star 2) indicate higher N over Si utilization (N>Si). The rectangle indicates the total**
**range of possible isotopic values. The respective fractionation factors are given in red.**



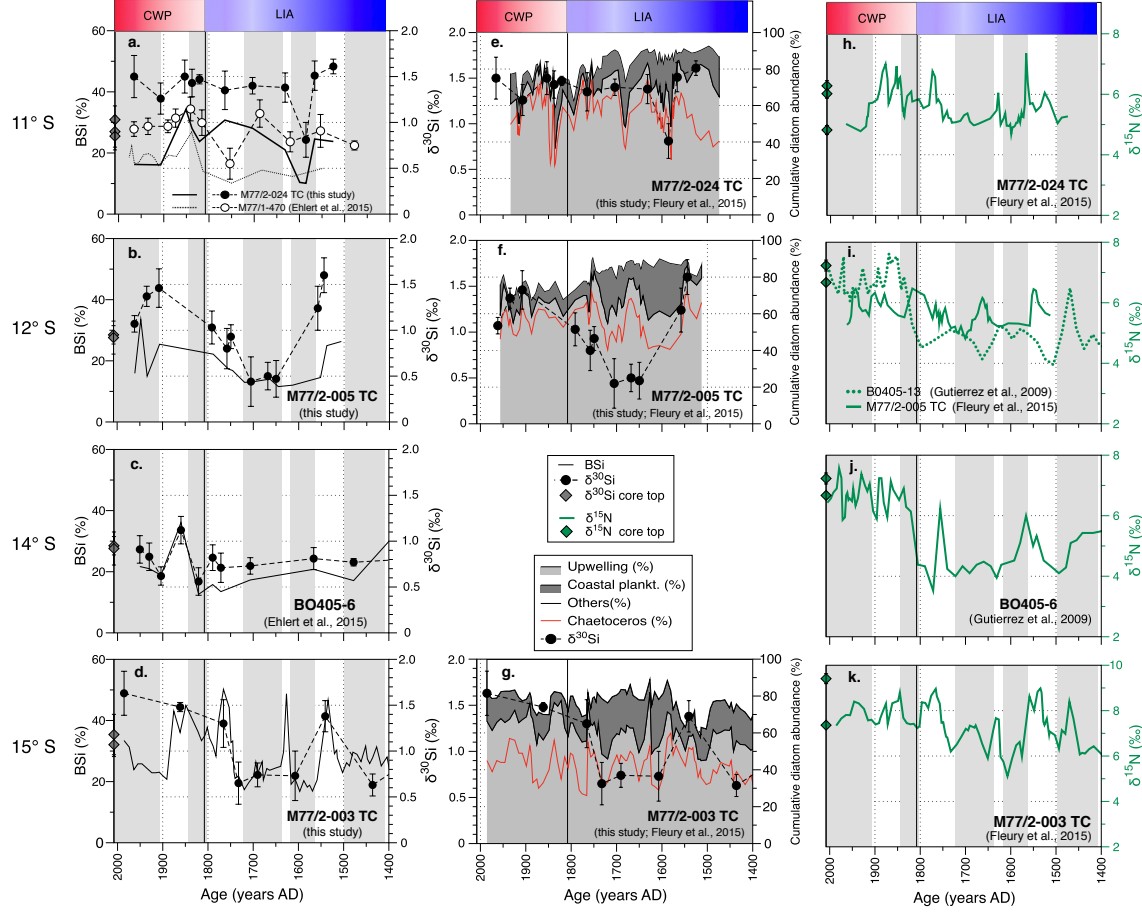


Figure 4: Downcore records of BSi (wt%), δ30SiBSi (‰, 2 SD error bar of repeated sample measurements) records of cores: (a) M77/2-024-5TC and, M77/1-470 (Ehlert et al., 2015) (b) M77/2-005-3TC (c) BO405-6 (Ehlert et al., 2015; Gutiérrez et al., 2009) and (d) M77/2-003-2TC. The cumulative diatom assemblages are compared to δ30Si for core e.) M77/2-024-5TC, f.) M77/2-005-3TC and g.) M77/2-003-2TC: Upwelling species - light gray; Coastal planktonic – gray; Other species – white; *Chaetoceros* sp. – red dashed line; δ30SiBSi – black dots; the black line indicates the transition between the LIA and the CWP. For comparison previously published δ15Nbulk (‰) are shown for cores h.) M77/2-024-5TC (Fleury et al., 2015), i.) M77/2-005-3TC (Fleury et al., 2015) and BO405-13 (Gutiérrez et al., 2009), j.) BO405-6 (Gutiérrez et al., 2009) and k.) M77/2-003-2TC (Fleury et al., 2015). All records are sorted by latitude from top (11°S) to bottom (15°S). The time intervals for the CWP (red) and the LIA (blue) are highlighted in (a); the horizontal grey shading indicates humid periods (Fleury et al., 2015).

857

858

859

**Figure 5: (a)** Direct comparison of $\delta^{15}N_{bulk}$ versus $\delta^{30}Si_{BSi}$ for modern surface sediments (modified from Ehlert et al., 2015): The dashed lines indicate 1:1 utilization of different $\delta^{15}NO_3^-$ source values (7‰, 7.9‰, 8.35‰ and 11.3‰) between 9°S and 15°S (based on Mollier-Vogel et al., 2012), the rectangle marks the respective range of isotope values that can be expected in sediment samples for nutrient utilization with source values of 1.5‰ ($\delta^{30}Si(OH)_4$) and 8.35‰ ($\delta^{15}NO_3^-$); (inset) Schematic overview of nutrient utilization associated with changes in the isotopic compositions of both $\delta^{15}N$ and $\delta^{30}Si$: the black star marks the source signature (or 100% utilization) for $\delta^{15}N$ and for $\delta^{30}Si$, the grey star marks the respective isotopic compositions for 0% utilization, the black dashed line indicates the 1:1 utilization for $NO_3^-$:$Si(OH)_4$, respectively. Ratios that plot above the utilization lines reflect $Si(OH)_4$ limitation, as indicated by the dark grey and light grey dotted line representing ratios of 1:2, whereas data points below record stronger $NO_3^-$ limitation, as indicated by the dark grey and light grey dashed lines representing ratios of 2:1 and 15:1. The rectangle indicates the total range of possible isotopic values. **(b-d)** Downcore comparison of $\delta^{15}N_{bulk}$ and $\delta^{30}Si_{BSi}$ for cores 024TC (diamonds), 005TC (grey stars), 003TC (grey squares) and BO405-6 (grey triangles; Ehlert et al., 2015), for the CWP and the LIA. For the LIA the sample values are separated into arid (d) and humid periods (b).

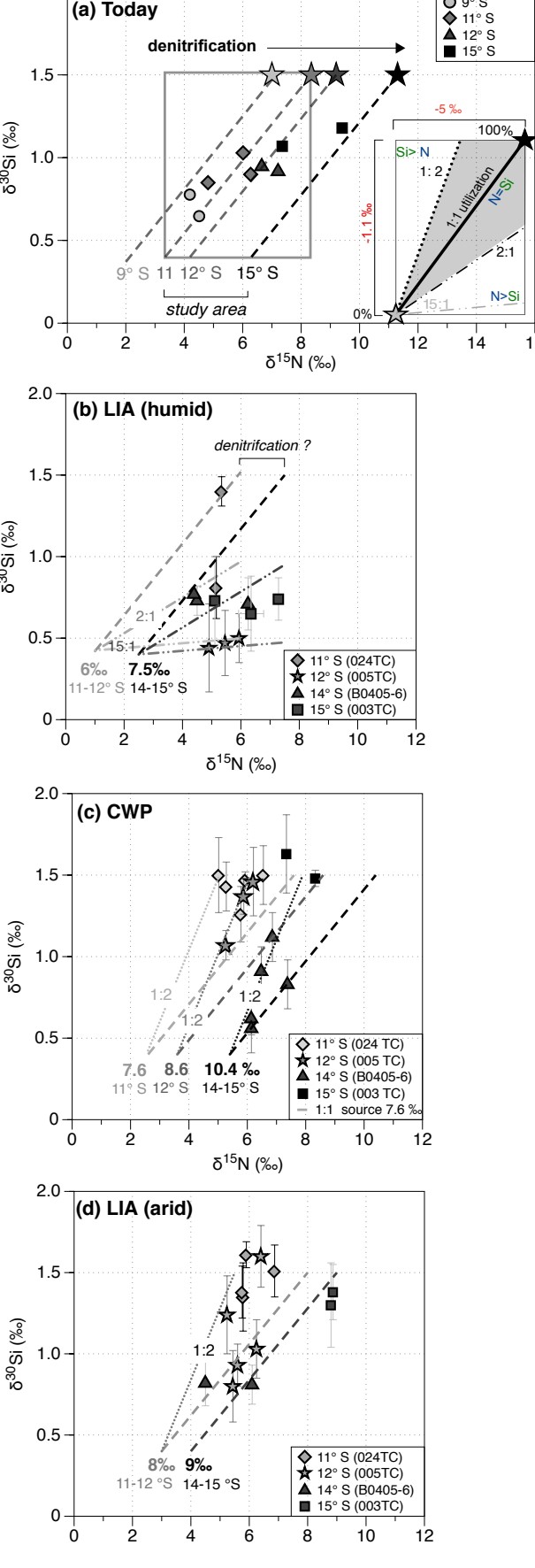

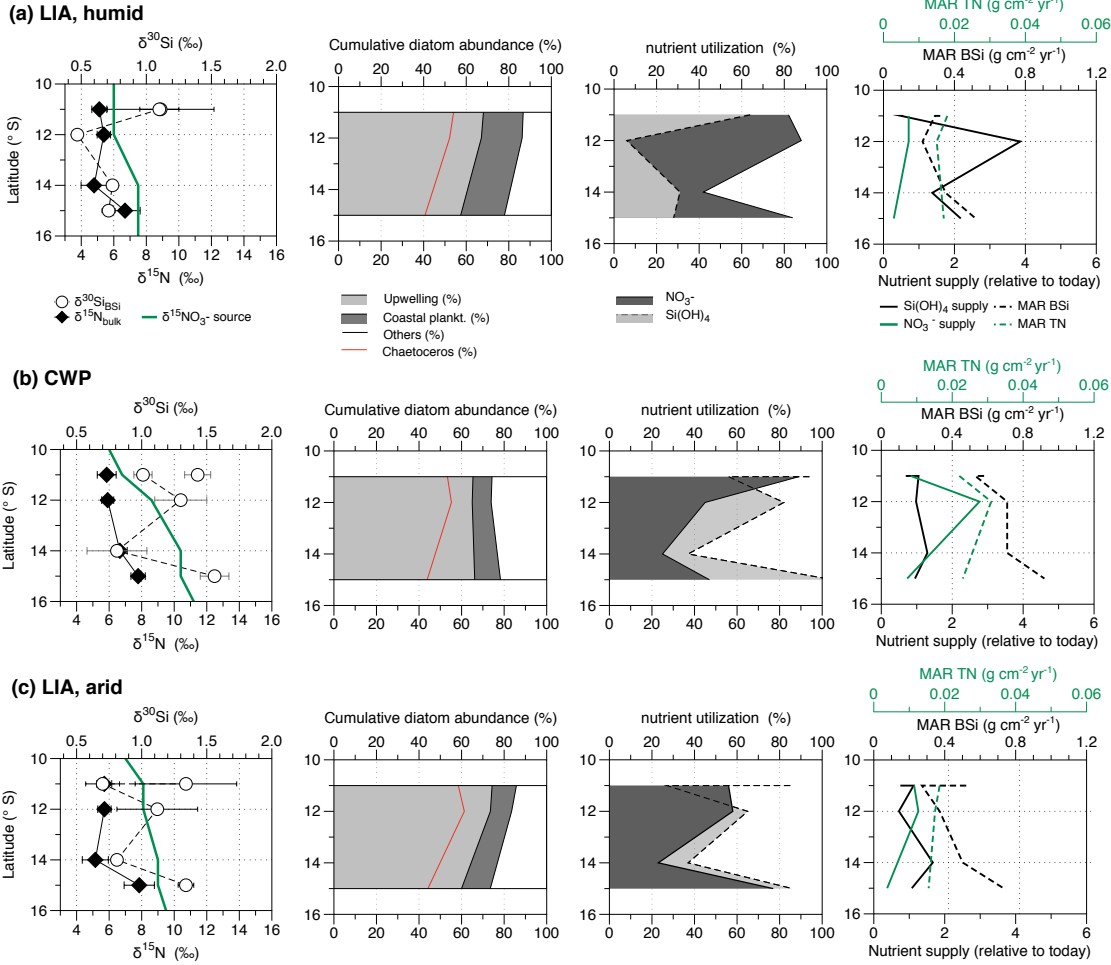

**Figure 6: Latitudinal comparison of (from left to right) mean δ15Nbulk (‰, black diamonds) and δ30SiBSi (‰, white circles) and the calculated δ15NO3- source values (green line), the mean cumulative diatom abundance (%; calculated from Fleury et al., 2015), the respective nutrient utilization of NO3- (grey, solid line) and Si(OH)4 (dashed area and line), and MAR TN (g cm-2 yr-1) and MAR BSi (g cm-2 yr-1) together with nutrient supply relative to today for (a) the humid phases of the LIA, (b) the CWP and (c) the arid phases of the LIA and . Please note that for δ15Nbulk values the mean was calculated for all available values for each time period and not only for samples, for which also δ30SiBSi values are available. Error bars mark the 1 SD of the mean values.**

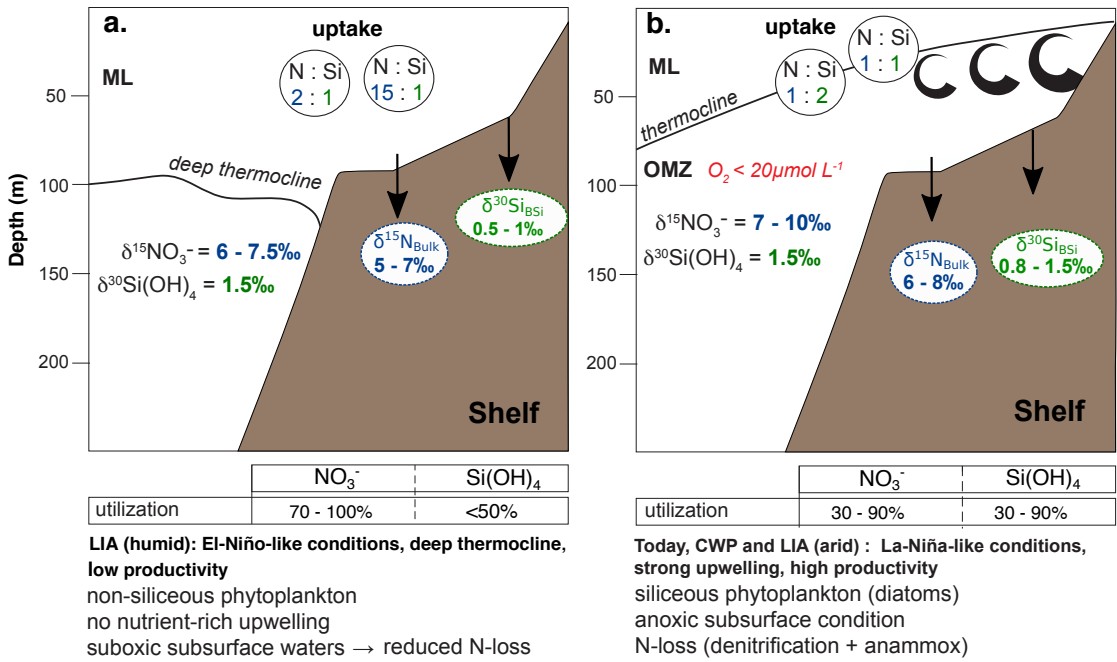

**Figure 7: Schematic nutrient (Si(OH)₄ and NO₃⁻) cycle models for the Peruvian mixed layer (ML) and Oxygen Minimum Zone (OMZ) along the shelf area (0-200m water depth) during the last 600 years. The NO₃⁻:Si(OH)₄ (N:Si) indicates the ratio in which both nutrients are taken up during biological production in surface waters.**