# Peer review of "Latitudinal variations of $\delta^{30}$ Si and $\delta^{15}$ N signatures along"

_Biogeosciences, 2018_

## Referee Comment (RC1) · Anonymous Referee #1 · 9 May 2018

Latitudinal variations of d30Si and d15N signatures along the Peruvian shelf: quantifying the effects of nutrients utilization versus denitrification over the past 600 years

Doering et al.

The authors present some new diatom d30Si records from the last few centuries from three cores off the Peruvian upwelling margin. These recent records are hard to come by, and the data are of good quality – also, the combination of different isotope systems to probe the decoupling of N and Si cycles is an interesting and useful approach (and

one that I think is underused in the community). The paper is generally written well, and structured clearly. As such, I think that these records should be published, and Biogeosciences is an appropriate outlet.

However, I have a few reservations about the (potential over-) interpretation of the data in places, I would like to see some additional detail in the methods, and I have a few additional minor comments and suggestions.

1. Interpretation of data:

I really like the novel cross-plot approach used in this manuscript, comparing the relative changes between silicon and nitrogen isotopes. I find this approach more convincing than some of the descriptions of the downcore variations in each isotope system individually. Also I wonder to what extent the data available can substantiate the conclusions. I think the novel approach is worthy of publication in Biogeosciences, and this study is a good illustration of what is possible, but I also think that the data description needs to be clarified, and that there are some caveats in the discussion should be included.

In section 3.1. I'm not very convinced by the descriptions – they don't seem to match up well with the plots in figure 3 to me. For example, the on line 193 say that between 12 and 15°S the d30Si have a mean lower value during the LIA than the CWP – however, this really isn't the case for B0405-6, and isn't thoroughly convincing for the other cores either. This statement also hides variability observed within the LIA. There are other examples of this throughout the section when referring to both d30Si and d15N. There are also examples of this in section 3.2.2 e.g. lines 379 onwards – at both 12 and 15°S there are d30Si values from the humid LIA that are the same as the modern values (if I've interpreted the grey horizontal bars on figure 3 correctly). Please make sure that your words fit the data.

The authors use -1.1 per mil as a fractionation factor, but there is, in fact, a large range in this fractionation factor. The authors use this value in their calculations (line 261)

but how does the uncertainty on this value influence the findings? Perhaps the authors could think about some sensitivity studies?

2. Methods:

There is no mention in the manuscript about the uncertainties that we have about the fractionation factor of silicon isotopes during uptake by diatoms (see comment below). It is possible that the downcore variations are driven by diatom species differences (I'm not saying that they are – it's just a possibility). This possibility can be readily dismissed by including information about downcore species differences. Ideally, diatom counts would be done on the separated and cleaned material (mentioned in lines 143 onwards). However, if this isn't possible at this stage (i.e. there is not cleaned material remaining), then perhaps the authors could at least plot their downcore isotope variations relative to the diatom abundance data mentioned on line 181 (Fleury et al., 2015)? This would at least give some indication of whether or not species changes are driving the isotope variations.

Lastly, there is no real mention in the methods section about how the sampling was carried out with respect to the fine-laminations (line 132). Were the samples taken from individual laminations? Was there any possibility of signal aliasing? Given the discussion about resolution in the manuscript later on (e.g. line 209), I think it would be valuable to clarify the sampling resolution upfront in the methods section.

3. Minor comments:

The title is appropriate for the contents of the paper. The abstract is a generally good, concise summary, although the authors should make it clear in the abstract that it's only some of the d30Si data that are new to this study. I didn't glean initially that the d15N data were published, and was confused to start with as to why there was no d15N methods section!

The references are generally good. However, in the introduction, the authors should at

least mention some of the caveats associated with diatom d30Si interpretation, namely the possibility pf species specific fractionation (e.g. Sutton et al., 2013) and dissolution (Demarest et al., 2009). See comments above regarding species specific fractionation; dissolution impacts on ïĄď30Si is more challenging to investigate as there isn't agreement in the literature about how big the dissolution signal might be (Egan et al., 2012; Wetzel et al., 2014) – however, I think at least a sentence should be included to note it as a possible complicating factor.

On line 91, the authors should be more specific than "high amounts" – are you referring to high concentrations, fluxes, or both?

What do the +/- signs on lines 97 onwards represent?

One line 114, the authors could add a few words to explain why the steady-state system is appropriate here. This arises again mater in the manuscript, but I think it would help to clarify the choice here as well.

On line 138, the authors should remove "in study of".

The sentence on line 178 is not complete – please rewrite. Also the short paragraph on line 186 onwards seems a little misplaced – I'd suggest the end of that section is rephrased.

Line 290: please avoid using "a bit lower" – rephrase.

Line 352: I'm not sure what you mean by "horizontal alignment" – could you please clarify?

Line 311: Is there no means of assessing changes in downcore phytoplankton assemblages, as a comparison to the modern data from Sanchez et al? Biomarkers?

Figure caption 4: The caption points towards figures c-e, when they should be figures b-d.

Figure 6: The fonts are too small in places.

Additional references:

Demarest, M.S., Brzezinski, M.A. and Beucher, C., 2009. Fractionation of silicon isotopes during biogenic silica dissolution. Geochimica et Cosmochimica Acta, 73: 5572-5583. Egan, K. et al., 2012. Diatom silicon isotopes as a proxy for silicic acid utilisation: A Southern Ocean core top calibration. Geochimica et Cosmochimica Acta, 96: 174-192. Sutton, J., Varela, D., Brzezinski, M.A. and Beucher, C., 2013. Species-dependent silicon isotope fractionation by marine diatoms. Geochimica et Cosmochimica Acta, 104: 300-309. Wetzel, F., de Souza, G. and Reynolds, B., 2014. What controls silicon isotope fractionation during dissolution of diatom opal? Geochimica et Cosmochimica Acta, 131: 128-137.

---

## Referee Comment (RC2) · Anonymous Referee #1 · 23 May 2018

Many thanks to the authors for correcting the plot. This is a great improvement, and has addressed my main concern.
* * *

---

## Short Comment (SC1) · 23 May 2018

Referee #1 is right that the description in section 3.1 is not matching figure 3 well. This unfortunately occured due to an error in figure 3, where the grey areas depicting humid conditions were shifted by 50 years indicating too young time periods. This was now corrected as can be found in the figure attached.

[Figure]

**Fig. 1.**

---

## Author Comment (AC1) · 4 Jun 2018

We highly appreciate the valuable comments and suggestions by the anonymous reviewer on our manuscript, especially the approval of our cross-plot approach to compare silicon and nitrogen isotope signatures. According to the reviewers suggestions we added more detailed information about factors potentially affecting the silicon isotope compositions in the introduction and methods section, namely the influence of a changing diatom assemblages and dissolution effects on the downcore record. Furthermore, one major comment addressed a discrepancy between the text in the results

and the discussion concerning figure 3. Some statements have been rephrased, however, the main issue was a mistake in figure 3, in which the grey bars indicating humid conditions were accidentally shifted by 50 years. The figure has been revised accordingly. In the following each comment is answered in detail in the order of the comments provided by the reviewer.

In section 3.1. I'm not very convinced by the descriptions – they don't seem to match up well with the plots in figure 3 to me. For example, the on line 193 say that between 12 and  $15^{\circ}$ S the d30Si have a mean lower value during the LIA than the CWP – however, this really isn't the case for B0405-6, and isn't thoroughly convincing for the other cores either.

The reviewer is right in that the actual mean LIA  $\delta$ 30SiBSi value for BO405-6 is not significantly different from the CWP. To highlight the individual changes in all cores we now present mean values and 2SD variability for 12, 14 and 15°S in Line 225-228: 'The  $\delta$ 30Si records follow a similar trend of lower mean  $\delta$ 30SiBSi values of 0.8  $\pm$  0.2‰ (2SD, 12°S), 0.8  $\pm$  0.1‰ (14°S) and 1  $\pm$  0.2‰ (15°S) during the LIA to more variable and higher mean values of 1.3  $\pm$  0.4‰ (12°S), 0.8  $\pm$  0.4‰ (14°S) and 1.5  $\pm$  0.2‰ (15°S) during the CWP.'

This statement also hides variability observed within the LIA. There are other examples of this throughout the section when referring to both d30Si and d15N. There are also examples of this in section 3.2.2 e.g. lines 379 onwards – at both 12 and 15°S there are d30Si values from the humid LIA that are the same as the modern values (if I've interpreted the grey horizontal bars on figure 3 correctly). Please make sure that your words fit the data

Thanks to the remarks of the reviewer we found that there was an error in Figure 3, in that the grey horizontal bars which mark the humid conditions throughout the last 600 years were accidentally displaced by 50 years. The figure was corrected accordingly and thus the text now matches the figure.
The authors use -1.1 per mil as a fractionation factor, but there is, in fact, a large range in this fractionation factor. The authors use this value in their calculations (line 261) but how does the uncertainty on this value influence the findings? Perhaps the authors could think about some sensitivity studies?

The changes in the diatom abundances are not large enough (10-20% changes maximum) to substantially affect the isotopic values and there is only the fractionation factor for Chaetoceros brevis (-2.1; a polar species, not resting spores) available which is on average significantly different from the -1.1% value generally assumed. When calculating the changes in  $\varepsilon$  following Doering et al., (2016), the potential effect of the Chaetoceros fractionation factor on our data is less than 5%. This information was added in Lines 311-317:'To evaluate the impact of changes in 30 $\varepsilon$  on the  $\delta$ 30Si signatures the potential influence of species-specific fractionation was tested based on the impact of a -2.1% enrichment factors of Chaetoceros brevis (Sutton et al., 2013). However, the estimated impact on past  $\delta$ 30SiBSi records due to a change in the amount of Chaetoceros sp. Present in the sediment was less than 5% for all cores (M77/2-024-5TC, 005-3TC and 003-2TC) and thus did not alter the assumed 30 $\varepsilon$  of -1.1% substantially (based on calculations presented in Doering et al., 2016; calculations not shown).'

2. Methods: There is no mention in the manuscript about the uncertainties that we have about the fractionation factor of silicon isotopes during uptake by diatoms (see comment below). It is possible that the downcore variations are driven by diatom species differences (I'm not saying that they are – it's just a possibility). This possibility can be readily dismissed by including information about downcore species differences. Ideally, diatom counts would be done on the separated and cleaned material (mentioned in lines 143 onwards). However, if this isn't possible at this stage (i.e. there is not cleaned material remaining), then perhaps the authors could at least plot their downcore isotope variations relative to the diatom abundance data mentioned on line 181 (Fleury et al., 2015)? This would at least give some indication of whether or not species
changes are driving the isotope variations.

As the reviewer suggested we now plotted the diatom abundances as provided by Fleury et al., (2015) versus our  $\delta$ 30Si data (added to fig.3). In the Methods section a paragraph was added to explain the diatom abundances, Line 180-193:

2.3 Diatom assemblage data Diatom analysis of cores M/7/2-024-5TC, 005-3TC and 003-2TC was published previously based on three slides per sample and counting of a minimum of 300 valves for each sample (for details see Fleury et al., 2015). The diatom abundances are presented here for five groups representing different environmental conditions: Upwelling species – Chaetoceros sp., Skeletonema costatum, Thalassionema nitzschioides var. nitzschioides; Coastal planktonic – Actinocyclus spp., Atinoptychus spp, Asteromphalus spp., and Coscinodiscus sp.; Other diatom species – Nitzschia spp., Rhizosolenia spp. and Thalassiosira spp., Cyclotella spp., Cocconeis sp.; The diatom assemblage abundance is compared to  $\delta$ 30SiBSi compositions for cores M77/2-024-5TC, 005-4TC and 003-2TC to investigate if changes in the assemblage have influenced the isotopic record. While diatom counts have been performed on bulk sediment samples ïĄd'30SiBSi was measured on the 11-32 $\mu$ m. However, it was shown previously that this size range closely resamples the main assemblage, which allows studying the influence of changes in the diatom assemblage on the  $\delta$ 30SiBSi record (Ehlert et al., 2012; 2013).

Additional text concerning the diatom assemblages has been added: Line 239-243: 'However, comparison with the cumulative diatom assemblage indicates overall little difference in the amount of upwelling and coastal planktonic diatom species between the LIA and the CWP at 11°S (Fig. 4), with intervals of reduced abundances of upwelling species of generally less than 50 years, much shorter than the 100 to 150 year intervals observed at 12 and 15°S.' Line 247-248: '..., as  $\delta$ 30SiBSi analysis do not cover all short events (~50 years) of reductions in the abundance of upwelling diatom species (Fig. 3f).' Line 260-268:'Furthermore, the diatom assemblages (Fig. 3f-h; based on Fleury et al., 2015) show a close correspondence of the amount of upwelling
species and  $\delta$ 30SiBSi signatures, with decreases of up to 20% in upwelling species often linked to a reduction of  $\delta$ 30SiBSi by about 0.5 -1‰However, not every decrease in  $\delta$ 30SiBSi is mirrored by a change in the diatom assemblage and vice versa (e.g. Fig. 3g at 350 yrs BP). Overall the diatom assemblage data indicates little changes in the mean conditions at 11° S (024-5TC) and a slight reduction of upwelling strength at 12° S and 15° S during the LIA in comparison to the CWP. The most distinct shift from lower abundance of upwelling species (~50%) to higher values during the CWP (~70%) is found at 15° S (003-2TC) in accordance with the strongest changes in BSi and  $\delta$ 30SiBSi at this location.

We further refer to the interpretation of the Doering et al., (2016), that the d30Si values are mostly affected by changes in the system off Peru, namely upwelling strength, and the associated diatom assemblage. This is emphasized now in the revised manuscript in response to the reviewers wishes, in Line 71-76: 'Accordingly, downcore records of  $\delta$ 30SiBSi off Peru are closely coupled to changes in the diatom assemblage with high signatures (>1‰ reflecting strong upwelling conditions and lower signatures (0-5-1‰ reflecting weak upwelling conditions (Doering et al., 2016). This coupling was previously shown to be mainly the consequence of changes in the abundance of different diatom groups during diatom succession linked to different upwelling strength (Doering et al., 2016) rather than potential species-specific fractionation (i.e. -0.5 to -2.1‰ *Suttonetal.*, 2013).'

Lastly, there is no real mention in the methods section about how the sampling was carried out with respect to the fine-laminations (line 132). Were the samples taken from individual laminations? Was there any possibility of signal aliasing? Given the discussion about resolution in the manuscript later on (e.g. line 209), I think it would be valuable to clarify the sampling resolution upfront in the methods section.

 $\delta$ 30Si and BSi measurements were generally performed on samples from 1 cm slices integrating several laminations. Only for core 003-2TC additional BSi measurements on material form single laminations was possible. For comparison data from lamina-
tions was interpolated to 1cm resolution. This information was added in Line 149 to 154 in the method section.: 'One cm slices of the sediment cores were sampled for BSi and silicon isotope measurements to ensure the availability of sufficient amount of diatoms for silicon isotope analysis (Tab. 1). For core 003-2TC additional BSi measurements on the extraction of sample material from individual laminations was possible (Fleury et al., 2015). As previously published  $\delta$ 15Nbulk values are based on samples from single laminations these were averaged to 1 cm resolution when directly compared to the  $\delta$ 30Si data in the following.'

3. Minor comments: The title is appropriate for the contents of the paper. The abstract is a generally good, concise summary, although the authors should make it clear in the abstract that it's only some of the d30Si data that are new to this study. I didn't glean initially that the d15N data were published, and was confused to start with as to why there was no d15N methods section!

It is clearly stated in line 22 to 23 in the abstract that we present three new  $\delta$ 30Si records and compare them to previously published  $\delta$ 30Si and  $\delta$ 15N records. Although we acknowledge the comment of the referee, we do not see the need to further emphasize this.

The references are generally good. However, in the introduction, the authors should at least mention some of the caveats associated with diatom d30Si interpretation, namely the possibility of species specific fractionation (e.g. Sutton et al., 2013) and dissolution (Demarest et al., 2009). See comments above regarding species specific fractionation; dissolution impacts on d30Si is more challenging to investigate as there isn't agreement in the literature about how big the dissolution signal might be (Egan et al., 2012; Wetzel et al., 2014) – however, I think at least a sentence should be included to note it as a possible complicating factor.

A sentence about potential biogenic opal dissolution has been added accordingly in in Line 42-45: While a potential fractionation of  $\delta$ 30Si signatures of biogenic opal during
dissolution of -0.55‰ has been reported previously (Demarest 2009), subsequent investigations of dissolution on diatom material from sediments have so far indicated that  $\delta$ 30Si signatures of diatoms within the sediments are generally unaffected by dissolution (Egan et al., 2012; Wetzel et al., 2014; Ehlert et al., 2016). On line 91, the authors should be more specific than "high amounts" – are you referring to high concentrations, fluxes, or both? We are referring to concentrations here, accordingly concentration values for Si(OH)4 and NO3- are now given in brackets. Line 101-102: 'Along the Peruvian margin the main source for the high amounts of upwelled nutrients (30  $\mu$ mol L-1 for both Si(OH)4 and NO3-; Bruland et al., 2005)'

What do the +/- signs on lines 97 onwards represent?

For the  $\delta$ 30Si the +/- always indicates the 2SD external reproducibility, only for the  $\delta$ 15NO3- values taken from Rafter 2012 the +/- indicates the 1SD variability of several water masses. This has now been added to the text in brackets in lines 108 and 109.

One line 114, the authors could add a few words to explain why the steady-state system is appropriate here. This arises again mater in the manuscript, but I think it would help to clarify the choice here as well.

Based on observations of Ehlert et al. 2012, the corresponding reference was added to Lines 125-126.

On line 138, the authors should remove "in study of".

'study of' was removed accordingly.

The sentence on line 178 is not complete – please rewrite. Also the short paragraph on line 186 onwards seems a little misplaced – I'd suggest the end of that section is rephrased.

The paragraph (Line 209-216) was rephrased as follows: 'This scenario is supported by a marked reduction in the concentrations of sedimentary redox sensitive trace metals such as molybdenum and rhenium (Salvatteci et al., 2014b; Sifeddine et al., 2008).
However, these conditions were not constant, instead short-term variations during both the LIA and the CWP are reflected for example mirrored by changes in diatom abundances, productivity sensitive element ratios (Br/Fe) and ïĄd'15Nbulk values (Fleury et al., 2015). These proxy records indicate multidecadal shifts between arid/humid conditions during the CWP and particularly the LIA during which marked short-term periods of arid conditions occurred (Fleury et al., 2015) (Fig. 3).'

Line 290: please avoid using "a bit lower" – rephrase.

The sentence was rephrased accordingly: Line 342: 'The shift towards a higher 1:2 NO3-:Si(OH)4 utilization during both the CWP and LIA (arid) indicates enhanced utilization of Si(OH)4 over NO3- leading to Si(OH)4 limitation as indicated by high Si(OH)4 utilization rates between 40% and 90%, and lower NO3- utilization rates between 25% and to 80% (Fig. 6b).'

Line 327: I'm not sure what you mean by "horizontal alignment" - could you please clarify?

The text (Line 377) was changed accordingly: 'horizontal alignment of the  $\delta$ 15Nbulk versus  $\delta$ 30SiBSi values'

Line 311: Is there no means of assessing changes in downcore phytoplankton assemblages, as a comparison to the modern data from Sanchez et al? Biomarkers?

It is possible to compare the diatom assemblages as done by Fleury et al., 2015. What we wanted to highlight here is that there are no modern  $\delta$ 15N and  $\delta$ 30Si isotope values available to compare modern El-Nino conditions with El-Nino-like conditions in the past as claimed for the LIA.

Figure caption 4: The caption points towards figures c-e, when they should be figures b-d.

The figure caption has been corrected accordingly.

BGD
Figure 6: The fonts are too small in places.

The fonts of the figure have been enlarged accordingly (see figure attached).

Additional references: Demarest, M.S., Brzezinski, M.A. and Beucher, C., 2009. Fractionation of silicon isotopes during biogenic silica dissolution. Geochimica et Cosmochimica Acta, 73: 5572-5583. Egan, K. et al., 2012. Diatom silicon isotopes as a proxy for silicic acid utilisation: A Southern Ocean core top calibration. Geochimica et Cosmochimica Acta, 96: 174-192. Sutton, J., Varela, D., Brzezinski, M.A. and Beucher, C., 2013. Species dependent silicon isotope fractionation by marine diatoms. Geochimica et Cosmochimica Acta, 104: 300-309. Wetzel, F., de Souza, G. and Reynolds, B., 2014. What controls silicon isotope fractionation during dissolution of diatom opal? Geochimica et Cosmochimica Acta, 131: 128-137.

The references have been included in the text and accordingly been added to the reference list.

Figure 3: Downcore records of BSi (wt%),  $\delta$ 30SiBSi (‰ 2 SD error bar of repeated sample measurements) and  $\delta$ 15Nbulk (‰ records of cores: (a) M77/1-470 (Ehlert et al., 2015), (b) M77/2-024-5TC, (c) M77/2-005-3TC and BO405-13 ( $\delta$ 15Nbulk Gutiérrez et al., 2009), (d) BO405-6 (Ehlert et al., 2015; Gutiérrez et al., 2009) and (e) M77/2-003-2TC. Records are sorted by latitude from left (11°S) to right (15°S). The time intervals for the CWP (red) and the LIA (blue) are highlighted in (a); the horizontal grey shading indicates humid periods (Fleury et al., 2015). The cumulative diatom assemblages are compared to  $\delta$ 30Si for core g.) M77/2-024-5TC, h.) M77/2-005-3TC and f.) ) M77/2-003-2TC: Upwelling species - light gray; Coastal planktonic – gray; Other species – white; Chaetoceros sp. – red dashed line;  $\delta$ 30SiBSi – black dots; the black line indicates the transition between the LIA and the CWP.

BGD

---

## Referee Comment (RC3) · Anonymous Referee #1 · 12 Jul 2018

I have read the authors' revised manuscript, and my comments and suggestions have been addressed well - I don't have anything further to add at this stage. This is a well-written paper that explores a new methodology for deconvolving the signals from different nutrient cycles, which is well-suited for Biogeosciences. I would suggest that the paper is published.

---

## Referee Comment (RC4) · P. Rafter (Referee) · 25 Jul 2018

By Patrick Rafter*

Summary The manuscript brings together new sedimentary d30S8 and (published) water column isotope measurements to examine the spatial and temporal variability of nutrient consumption in the Peruvian upwelling zone. This manuscript then applies a relationship between coretop sediment d30Si and d15N to estimate past changes in the source of upwelled nitrate d15N. This is an interesting application, but I have some

suggestions that I think are both necessary for supporting the core argument and will greatly improve the readability and therefore effectiveness of this manuscript.

Five suggestions to improve the manuscript First, I think the d30Si and d15N relationship in Figure 4A must be examined in a more robust manner. For example, I don't think it is appropriate to simply state that the surface sediment core measurements "remain close to the respective 1:1 utilization line" for nitrate to silicate in Figure 4A. This needs to be shown, regardless of this approach being previously published by Ehlert et al. (2015). Specifically, the study would be improved with a more precise quantification of this relationship in Figure 4A. Similarly, the frequent use of "correlation" in the text (n=9) is not supported with any statistics. This must be fixed.

Second, I think it would improve this study if the "nutrient utilization" plots in Figures 2 and 3 were separated instead of overtop each other. The estimated nitrate and silicate utilization by phytoplankton should be on separate plots.

Another strong recommendation is to compile the sediment core measurements per latitude in Figure 3 in a more comprehensible way. The reader will have a much easier time understanding the relationships in Figure 3 if they: (1) Show one measurement at a time and (2) Are "stacked" from low to high southern latitudes. In this new figure, the bulk sediment d15N for all sites will be shown 'stacked' in one column that moves from 11°S at the top to 15°S at the bottom. This new arrangement of the Figure 3 data will include the same data, but in a more easily understood arrangement and will allow the reader to identify the spatial and temporal variability of each proxy.

Fourth, the use of sedimentary percentages is not appropriate; we must see the percentage data expressed as a mass accumulation rate or MAR. This needs to be changed before I would have confidence in the interpretation (any interpretation) of the percentage measurements. The addition of MARs may actually improve the interpretation of the data, since there seems to be some confusion in the interpretation of these measurements (see Lines 204 and on).

Finally, I don't think the Discussion section is the location for describing every individual wiggle of the observations and /or the estimated source nitrate d15N. I think that an improved manuscript would have a robust statistical examination of the surface sediment d15N and d30Si (from Figure 4) in the Results section followed by text that describes the temporal variability of Figures 4 and 5. In this way, the Discussion section can be used to discuss the observed / estimated spatial and temporal changes, which will be more easily understood and (I think) enjoyable for the reader. No one wants to read a listing of which way the wiggles are wiggling and when. :)

Line-by-line notes: Figure 2: Illustrating the nutrient utilization in this way is not helpful. I would discard this plot and explore other options.

Figure 3: As I mentioned above, these plots are almost impossible for the casual reader to understand. I would re-plot using one observation at a time. Furthermore, they should move from lowest latitude at the top to highest latitude at the bottom (this is intuitive).

Figure 4: Incorrectly labeled as Figures A and C-E in caption.

Figure 5: Once again, the nutrient utilization plot is painful to look at and I cannot interpret it. Re-format. I also suggest that the percent sedimentary concentrations be converted to Mass Accumulation Rates.

Figure 6: This is probably the most useful of the figures.

Line 28: I don't think "humid conditions" is appropriate at this point; it is not common knowledge, it is not supported with evidence, and is out of place in the Abstract.

41: The isotope effects of nitrate assimilation were estimated at sites across the Pacific basin in Rafter and Sigman 2016. This is blatant self-promotion, but it is entirely relevant when citing reasonable isotope effects in the tropical Pacific (we found an average of 6 per mil). We also identify the origin of variability between Rayleigh (closed system) versus Open System isotopic fractionation.

62: This is a good sentence.

72: The statement that sedimentary d15N and d30Si variability "mainly depends on the degree of utilization of both nutrients" is not correct. This statements is disproven by this study's own Figure 4 and text later in the manuscript. This text should be restated.

81: remove "been"

89: remove "too".

99: When dealing with nitrate d15N, a difference in source waters of 5.5 to 7.0 per mil is quite large. Especially considering the small spatial difference between these sources.

139: The use of "BP" in age models is universally acknowledged to refer to the year 1950 of the Common Era. The date of BP cannot be adjusted to a new year. This will inevitably lead to much confusion.

201: Sedimentary concentration measurements are common, but they are not useful or appropriate proxies in 2018. MASS ACCUMULATION RATES are necessary. Apply and then we can re-examine the records.

273: correlation must be associated with an examination of the statistics.

*Why I am signing all my reviews Full transparency of peer reviews makes reviewers accountable for their work. I say this based on my own experience; my signed reviews are more thoughtful and useful, which leads to better science.

---

## Author Comment (AC2) · 24 Sep 2018

Response to the reviewers comments

We thank the reviewer (Patrick Rafter) for his helpful suggestions to improve the manuscript. According to his comments we restructured the manuscript and established separate Results and Discussion sections. Further, all equations have now been included in the Methods sections. For Figure 4a a more detailed and comprehensive description and explanation was added to the main text. Also, we added the

linear regression equations and respective correlation coefficient (r2) for figure 4 in a new supplementary figure S1. In the following each comment is answered in detail in the order of the comments provided by the reviewer. The changes were added to the previous changes applied to the manuscript following suggestions of reviewer 1.

In the following references for lines are always given for the revised manuscript version (not shown).

First, I think the d30Si and d15N relationship in Figure 4A must be examined in a more robust manner. For example, I don't think it is appropriate to simply state that the surface sediment core measurements "remain close to the respective 1:1 utilization line" for nitrate to silicate in Figure 4A. This needs to be shown, regardless of this approach being previously published by Ehlert et al. (2015). Specifically, the study would be improved with a more precise quantification of this relationship in Figure 4A. Similarly, the frequent use of "correlation" in the text (n=9) is not supported with any statistics. This must be fixed. We added a more comprehensive explanation of what is shown in Fig. 4a and how it relates to our results in S1 and Fig 4 b-d. To address this issue we improved the text explaining Fig. 4 a and the conclusions we draw from it. This point will also be further addressed below, under the last point concerning the restructuring of the Results and Discussion sections.

This point will be also further addressed below, under the last point concerning the restructuring of the Results and discussion.

The improper use of the word "correlation" was checked throughout the whole manuscript. See lines: Lines 25, 99, 572, 574 and 575, as well as in the figure captions.

Second, I think it would improve this study if the "nutrient utilization" plots in Figures 2 and 3 were separated instead of overtop each other. The estimated nitrate and silicate utilization by phytoplankton should be on separate plots. Following the reviewer's suggestion we changed figure 2 to improve the visibility of the nutrient utilization. However we did not separate NO3- and Si(OH)4 utilization given that the purpose of this figure

is exactly to illustrate the relationship of the changes in both systems. See revised Figures 2 below.

Another strong recommendation is to compile the sediment core measurements per latitude in Figure 3 in a more comprehensible way. The reader will have a much easier time understanding the relationships in Figure 3 if they: (1) Show one measurement at a time and (2) Are "stacked" from low to high southern latitudes. In this new figure, the bulk sediment d15N for all sites will be shown 'stacked' in one column that moves from 11°S at the top to 15°S at the bottom. This new arrangement of the Figure 3 data will include the same data, but in a more easily understood arrangement and will allow the reader to identify the spatial and temporal variability of each proxy. Figure 3 was modified and we now separated the different proxies (see below). Furthermore, the records are now sorted from 11 to 15°S from top to bottom.

Fourth, the use of sedimentary percentages is not appropriate; we must see the percentage data expressed as a mass accumulation rate or MAR. This needs to be changed before I would have confidence in the interpretation (any interpretation) of the percentage measurements. The addition of MARs may actually improve the interpretation of the data, since there seems to be some confusion in the interpretation of these measurements (see Lines 204 and on). For presentation of the BSi and TN data in Figures 2 and 5, we now calculated the respective accumulation rates based on data presented by Gutierrez et al., 2009 given for 12°S and 14°S, respectively, as there are no accumulation rates available for cores M77/2-24, 005 and 003. As the BSi (wt%) data is new we still show these values in Figure 3. However, we want to point out that there is no confusion about the interpretation of BSi data. As visible in the comparison of the records of core M77/2-24 with the previously published core M77/1-470 from the same latitude the BSi (wt%) is overlapping during the CWP, but BSi values of core 24 remain high throughout the LIA and d30Si signatures of core 24 are higher by about 0.5‰ at all times. From line 204 onwards it is discussed how this difference in d30Si can occur in the 2 records located so close to each other.

[Figure]

Text added: Lines 169-174: Unfortunately, no material was left of the cores studied to estimate dry bulk densities to calculate mass accumulation rates (MAR), therefore values were used from cores BO413 (12 °S) and BO406 (14 °S; Gutierréz et al. 2009) which were generally close to 0.02 (g cm-2 yr-1) during the LIA and 0.03 (g cm-2 yr-1) during the CWP. The exact bulk MAR values (g cm-2 yr-1) for each time period were multiplied by the fractional concentration of BSi and TN (Fleury et al., 2015) to calculate the MAR BSi and MAR TN (Figs. 2 and 5).

Finally, I don't think the Discussion section is the location for describing every individual wiggle of the observations and /or the estimated source nitrate d15N. I think that an improved manuscript would have a robust statistical examination of the surface sediment d15N and d30Si (from Figure 4) in the Results section followed by text that describes the temporal variability of Figures 4 and 5. In this way, the Discussion section can be used to discuss the observed / estimated spatial and temporal changes, which will be more easily understood and (I think) enjoyable for the reader. No one wants to read a listing of which way the wiggles are wiggling and when. Following the suggestion of the reviewer the Results and Discussion section was restructured i.e. split. The Equation for Nutrient utilization was moved to the Methods section (new section 2.4; Lines 208-227). To address the statistical examination we added a supplementary figure (see below) that gives all linear regressions and r2 used for the calculations of the d15NO3-source signatures. For the surface sediment samples a statistical examination is of little gain due to the small sample amount concerned, however we added a more detailed explanation about the information gained by Fig. 4a that should resolve this issue.

Lines 238-264:' Based on modern observations from the water column it is known that NO3- and Si(OH)4 are incorporated in a 1:1 ratio when diatoms dominate the phytoplankton assemblage (Brzezinski, 1985; Ragueneau et al., 2000) this ratio, however, can vary between 2:1 and 1:2 on the shallow Peruvian shelf (Grasse et al., 2016). Assuming this is true for the past as well, a change in relative uptake will be visible in the d15NBulk versus d30SiBSi relationship in surface sediments. We therefore applied a direct comparison of both isotope signatures in the surface sediments, following an approach of Ehlert et al. (2015). To evaluate how modern sediments record the relative nutrient utilization of both NO3- and Si(OH)4, surface sediment values were sorted by latitude (9, 11, 12 and 15°S) and compared to their respective d15NO3- and d30Si(OH)4 source and utilization relationship at each location. First, based on the known d30Si(OH)4 and NO3- source signatures of subsurface waters we can calculate a theoretical utilization assuming 1:1 incorporation ratios of both Si(OH)4 and NO3-. There are several factors that may cause d15N:d30Si ratios to deviate from this 1:1 utilization uptake ratio of both nutrients: 1) under iron-limitation more Si(OH)4 relative to NO3- will be taken up resulting in heavier silicified diatoms and shifting to a 1:4 N:Si uptake ratio (Hutchins and Bruland, 1998; Franck et al., 2000); 2) under prevalence of non-siliceous phytoplankton groups more NO3- than Si(OH)4 is incorporated (Conley and Malone, 1992; Wilkerson and Dugdale, 1996), the ratio might shift up to 15:1 (Grasse et al., 2016). Therefore, the deviation from the 1:1 NO3- to Si(OH)4 ratio can serve as an indicator for the degree of relative utilization of NO3- over Si(OH)4 (Grasse et al., 2016; see utilization schematic indicated in Figure 4a). The surface samples from 9°S (Fig. 4) indicate low to moderate utilization of both nutrients. The sample values, however, plot beneath the 1:1 utilization line, reflecting higher NO3- than Si(OH)4 utilization, which is supported by the calculated NO3- and Si(OH)4 utilization as based on a steady state calculation (Fig. 2b). The data from 11°S (Fig. 4a) all plot close to the middle of the respective 1:1 utilization line at about the middle indicating about 50% utilization for both NO3- and Si(OH)4, agreeing with the calculated utilization values shown in Fig. 2b. For 15° S the sample values plot further to the left of the 1:1 utilization, indicating higher utilization of Si(OH)4 than NO3- (Fig. 4a, 2b). This displacement to the left supports a NO3-:Si(OH)4 ratio of 1:2 or 1:4, which agrees with direct observations from the surface waters in areas of strong upwelling (Grasse et al., 2016).'

Further, the calculation of the d15NO3- source based on linear regression was also moved to the Methods (now section 2.4; Lines 266-274). This paragraph was also

rephrased accordingly: Lines 257-265:' For the past time periods of the CWP and LIA conditions are thought to shift between two states (1) high upwelling intensity, high productivity and strong N-loss processes and (2) low productivity and N-loss intensity. For either state we assume that The linear relationship between d30SiBSi and d15Nbulk signatures should thus also be a direct indicator of the d15NO3- source signatures. in the past during the CWP and the LIA. Depending on the exact conditions the uptake of NO3- to Si(OH)4 might be in a ratio of for example 1:1, 2:1 or 1:2. If the conditions remain stable over the time periods samples should represent a linear relationship similar to the utilization lines indicated in Fig. 4. Assuming such a linear relationship of d15Nbulk to d30SiBSi samples for past periods, it is possible to first calculate the respective d15NO3- source values and then further quantify the amount of NO3- utilization.'

Lines 291-294:'For all time periods and latitudes, the linear regressions as well as correlation coefficient (r2) are given in the supplements (Fig. S1). The results are presented in the following as the resulting d15NO3- source values and the theoretical ratio of nutrient utilization (i.e. 1:1 or 2:1, 15:1, etc.; Fig. 4. b-d) for each latitudinal range to compare the latitudinal trends between the CWP and LIA'

Additionally, paragraphs within the previous Results and Discussion section mainly describing results were taken omitted (see Lines 424-433; 458-466; 522-535) or rephrased (Line 467-471) and a separate Results section was written (subsections 3.1 Biogenic opal and silicon isotope signatures and 3.2 d15NO3- source signatures, nutrient utilization and supply) see lines 336- 362.

Line 29: I don't think "humid conditions" is appropriate at this point; it is not common knowledge, it is not supported with evidence, and is out of place in the Abstract. 'humid conditions' has been excluded from this sentence as suggested by the reviewer.

The isotope effects of nitrate assimilation were estimated at sites across the Pacific basin in Rafter and Sigman 2016. This is blatant self-promotion, but it is entirely rele-

vant when citing reasonable isotope effects in the tropical Pacific (we found an average of 6 per mil). We also identify the origin of variability between Rayleigh (closed system) versus Open System isotopic fractionation. The given fractionation factors actually match values calculated for the Peruvian shelf, the appropriate references were added accordingly. Line 45: .... , which agree well with estimates for the Peruvian shelf (Ehlert et al., 2012; Mollier-Vogel et al., 2012; Grasse et al., 2016). However, the reference to Rafter and Sigman 2016 was added in Line 116, as it also gives a detailed overview about the d15NO3- values within the EUC.

Line 81: remove "been" Line 98 'been' has been removed

89: remove "too" Line 106 (previously 89) 'to' has been removed accordingly

99: When dealing with nitrate d15N, a difference in source waters of 5.5 to 7.0 per mil is quite large. Especially considering the small spatial difference between these sources. Line 117: As pointed out by the reviewer a 1.5 per mil difference is not 'slight', therefore we removed 'slightly' from the sentence and replaced it with 'about 1.5‰:

201: Sedimentary concentration measurements are common, but they are not useful or appropriate proxies in 2018. MASS ACCUMULATION RATES are necessary. Apply and then we can re-examine the records. As now stated in line 169 and also line 303, there are unfortunately no accumulation rates available for the cores, as no dry bulk density was analyzed to calculate MAR for cores M772-024TC, 005TC and 003TC. Therefore, we used accumulation rates reported of cores BO406-13 and BO406-5 reported for the last 600 years by Guiterréz et al. 2009 to estimate the MAR TN and MAR BSi values in our cores. These data is shown in Figures 2 and 5 now. We left the actual BSi values in Figure 3, also as a comparison of the values with core M77/1-470. As accumulation rates mainly decrease during the LIA in comparison to the CWP, there should be little difference in the comparison of BSI and AR BSi concerning these two cores.

See also Line 169- 174:' Unfortunately, no material was left of the cores studied to

estimate dry bulk densities to calculate mass accumulation rates (MAR), therefore values were used from cores BO413 (12 °S) and BO406 (14 °S; Gutierréz et al. 2009) which were generally close to 0.02 (g cm-2 yr-1) during the LIA and 0.03 (g cm-2 yr-1) during the CWP. The exact bulk MAR values (g cm-2 yr-1) for each time period were multiplied by the fractional concentration of BSi and TN (Fleury et al., 2015) to calculate the MAR BSi and MAR TN (Figs. 2 and 5).'

Please also note the supplement to this comment:
https://www.biogeosciences-discuss.net/bg-2018-118/bg-2018-118-AC2-supplement.pdf
* * *
[Figure]

[Figure]

**Fig. 1.** Figure 2 (revised)

**Fig. 2.** Figure 3 (revised)

[Figure]

Fig. 3. Figure 5 (revised)

**Supplement:**

[Figure]

Figure S1: Calculation of $\delta^{15}NO_3^-$ source signatures based on equations of the linear regression gained by the direct comparison of $\delta^{15}N_{Bulk}$ versus $\delta^{30}Si_{BSi}$ for the CWP (a-c), the LIA (arid; d, e) and LIA (humid). Black lines indicate the estimated linear regression. Dashed lines indicate the resulting 1:1 utilization line.

---

## Author Response (AR1)

**Response to the reviews**

Please find in the following a point-to-point response to the reviews, a list of the relevant changes in the manuscript and the revised manuscript version with mark ups.

Please note that all line references given in the following refer to the manuscript with mark ups.

**Review #1**

In section 3.1. I'm not very convinced by the descriptions – they don't seem to match up well with the plots in figure 3 to me. For example, the on line 193 say that between 12 and 15°S the d30Si have a mean lower value during the LIA than the CWP – however, this really isn't the case for B0405-6, and isn't thoroughly convincing for the other cores either.

The reviewer is right in that the actual mean LIA $\delta^{30}Si_{BSi}$ value for BO405-6 is not significantly different from the CWP. To highlight the individual changes in all cores we now present mean values and 2SD variability for 12, 14 and 15°S in Line 315-317: 'The $\delta^{30}Si$ records follow a similar trend of lower mean $\delta^{30}Si_{BSi}$ values of 0.8 ± 0.2‰ (2SD, 12°S), 0.8 ± 0.1‰ (14°S) and 1 ± 0.2‰ (15°S) during the LIA to more variable and higher mean values of 1.3 ± 0.4‰ (12°S), 0.8 ± 0.4‰ (14°S) and 1.5 ± 0.2‰ (15°S) during the CWP.'

This statement also hides variability observed within the LIA. There are other examples of this throughout the section when referring to both d30Si and d15N. There are also examples of this in section 3.2.2 e.g. lines 379 onwards – at both 12 and 15°S there are d30Si values from the humid LIA that are the same as the modern values (if I've interpreted the grey horizontal bars on figure 3 correctly). Please make sure that your words fit the data

Thanks to the remarks of the reviewer we found that there was an error in Figure 3, in that the grey horizontal bars which mark the humid conditions throughout the last 600 years were accidentally displaced by 50 years. The figure was corrected accordingly and thus the text now matches the figure.

The authors use -1.1 per mil as a fractionation factor, but there is, in fact, a large range in this fractionation factor. The authors use this value in their calculations (line 261) but how does the uncertainty on this value influence the findings? Perhaps the authors could think about some sensitivity studies?

The changes in the diatom abundances are not large enough (10-20% changes maximum) to substantially affect the isotopic values and there is only the fractionation factor for Chaetoceros brevis (-2.1; a polar species, not resting spores) available which is on average significantly different from the -1.1‰ value generally assumed. When calculating the changes in ε following Doering et al., (2016), the potential effect of the Chaetoceros fractionation factor on our data is less than 5%. This information was added in Lines 220-225:

'To evaluate the impact of changes in $^{30}\varepsilon$ on the $\delta^{30}Si$ signatures the potential influence of species-specific fractionation was tested based on the impact of a -2.1‰ enrichment factors of *Chaetoceros brevis* (Sutton et al., 2013). However, the estimated impact on past $\delta^{30}Si_{BSi}$ records due to a change in the amount of *Chaetoceros* sp. Present in the sediment was less than 5% for all cores (M77/2-024-5TC, 005-3TC and 003-2TC) and thus did not alter the assumed $^{30}\varepsilon$ of -1.1‰ substantially (based on calculations presented in Doering et al., 2016; calculations not shown).'

2. Methods:

There is no mention in the manuscript about the uncertainties that we have about the fractionation factor of silicon isotopes during uptake by diatoms (see comment below). It is possible that the downcore variations are driven by diatom species differences (I'm not saying that they are – it's just a possibility). This possibility can be readily dismissed by including information about downcore species differences. Ideally, diatom counts would be done on the separated and cleaned material (mentioned in lines 143 onwards). However, if this isn't possible at this stage (i.e. there is not cleaned material remaining), then perhaps the authors could at least plot their downcore isotope variations relative to the diatom abundance data mentioned on line 181 (Fleury et al., 2015)? This would at least give some indication of whether or not species changes are driving the isotope variations.

As the reviewer suggested we now plotted the diatom abundances as provided by Fleury et al., (2015) versus our $\delta^{30}$Si data (added to fig.3). In the Methods section a paragraph was added to explain the diatom abundances, Line 193-206:

2.3 Diatom assemblage data
Diatom analysis of cores M77/2-024-5TC, 005-3TC and 003-2TC were published previously based on three slides per sample and counting of a minimum of 300 valves for each sample (for details see Fleury et al., 2015). The diatom abundances are presented here for three groups representing different environmental conditions (Fig. 3): Upwelling species – Chaetoceros sp., Skeletonema costatum, Thalassionema nitzschioides var. nitzschioides; Coastal planktonic – Actinocyclus spp., Atinoptychus spp, Asteromphalus spp., and Coscinodiscus sp.; Other diatom species – Nitzschia spp., Rhizosolenia spp. and Thalassiosira spp., Cyclotella spp., Cocconeis sp.;
The diatom assemblage abundance is compared to $\delta^{30}$Si$_{BSi}$ compositions for cores M77/2-024-5TC, 005-4TC and 003-2TC to investigate if changes in the assemblage have influenced the isotopic record. While diatom counts have been performed on bulk sediment samples $\delta^{30}$Si$_{BSi}$ was measured on the 11-32µm size fraction. However, it was shown previously that this size range closely resamples the main assemblage, which allows studying the influence of changes in the diatom assemblage on the $\delta^{30}$Si$_{BSi}$ record (Ehlert et al., 2012; 2013).

Additional text concerning the diatom assemblages has been added:
Line 402-406: 'However, comparison with the cumulative diatom assemblage indicates overall little difference in the amount of upwelling and coastal planktonic diatom species between the LIA and the CWP at 11°S (Fig. 4), with intervals of reduced abundances of upwelling species of generally less than 50 years, much shorter than the 100 to 150 year intervals observed at 12 and 15°S.'
Line 409-411: '…, as $\delta^{30}$Si$_{BSi}$ analysis do not cover all short events (~50 years) of reductions in the abundance of upwelling diatom species (Fig. 3f).'
Line 320-327-: The diatom assemblages (Fig. 3e-g; based on Fleury et al., 2015) show a strong association of the amount of upwelling species and $\delta^{30}$Si$_{BSi}$ signatures, with decreases of up to 20% in upwelling species often accompanied by a reduction of $\delta^{30}$Si$_{BSi}$ by about 0.5-1‰. However, not every decrease in $\delta^{30}$Si$_{BSi}$ is mirrored by a change in the diatom assemblage and vice versa (e.g. Fig. 3f at 1650 AD). Overall the diatom assemblage data indicates little changes in the mean conditions at 11°S (024-5TC) and a slight reduction of upwelling strength at 12°S and 15°S during the LIA in comparison to the CWP. The most distinct shift of lower abundances of upwelling species (~50%) to higher values during the CWP (~70%) is found at 15°S (003-2TC) corresponding to the strongest changes in BSi and $\delta^{30}$Si$_{BSi}$ at this location.'

We further refer to the interpretation of the Doering et al., (2016), that the $\delta^{30}$Si values are mostly affected by changes in the system off Peru, namely upwelling strength, and the associated diatom assemblage. This is emphasized now in the revised manuscript in response to the reviewers wishes, in Line 77-82:
'Accordingly, downcore records of $\delta^{30}$Si$_{BSi}$ off Peru are closely coupled to changes in the diatom assemblage with high signatures (>1‰) reflecting strong upwelling conditions and lower signatures (0-5-1‰) reflecting weak upwelling conditions (Doering et al., 2016). This coupling was previously shown to be mainly the consequence of changes in the abundance of different diatom groups during diatom succession linked to different upwelling strength (Doering et al., 2016) rather than potential species-specific fractionation (i.e. -0.5 to -2.1‰; Sutton et al., 2013).'

Lastly, there is no real mention in the methods section about how the sampling was carried out with respect to the fine-laminations (line 132). Were the samples taken from individual laminations? Was there any possibility of signal aliasing? Given the discussion about resolution in the manuscript later on (e.g. line 209), I think it would be valuable to clarify the sampling resolution upfront in the methods section.

$\delta^{30}$Si and BSi measurements were generally performed on samples from 1 cm slices integrating several laminations. Only for core 003-2TC additional BSi measurements on material form single laminations was possible. For comparison data from laminations was interpolated to 1cm resolution. This information was added in Line 158 to 163 in the method section.:

'One cm slices of the sediment cores were sampled for BSi and silicon isotope measurements to ensure the availability of sufficient amount of diatoms for silicon isotope analysis (Tab. 1). For core 003-2TC additional BSi measurements on the extraction of sample material from individual laminations was possible (Fleury et al., 2015). As previously published $\delta^{15}$N$_{bulk}$ values are based on samples from single laminations these were averaged to 1 cm resolution when directly compared to the $\delta^{30}$Si data in the following.'

3. Minor comments:

The title is appropriate for the contents of the paper. The abstract is a generally good, concise summary, although the authors should make it clear in the abstract that it's only some of the d30Si data that are new to this study. I didn't glean initially that the d15N data were published, and was confused to start with as to why there was no d15N methods section!

It is clearly stated in line 22 to 23 in the abstract that we present three new $\delta^{30}$Si records and compare them to previously published $\delta^{30}$Si and $\delta^{15}$N records. Although we acknowledge the comment of the referee, we do not see the need to further emphasize this.

The references are generally good. However, in the introduction, the authors should at least mention some of the caveats associated with diatom d30Si interpretation, namely the possibility of species specific fractionation (e.g. Sutton et al., 2013) and dissolution (Demarest et al., 2009). See comments above regarding species specific fractionation; dissolution impacts on d30Si is more challenging to investigate as there isn't agreement in the literature about how big the dissolution signal might be (Egan et al., 2012; Wetzel et al., 2014) – however, I think at least a sentence should be included to note it as a possible complicating factor.

A sentence about potential biogenic opal dissolution has been added accordingly in in Line 47-53:

While a potential fractionation of $\delta^{30}$Si signatures of biogenic opal during dissolution of -0.55‰ has been reported previously (Demarest 2009), subsequent investigations from the water column of the Southern Ocean did not see significant difference in the $\delta^{30}$Si values of material from the and $\delta^{30}$Si values in surface sediments is found (Varela et a., 2004; Fripiat et al., 2012; Closset et al., 2015). Furthermore, field studies and laboratory experiments based on sediments have so far indicated that $\delta^{30}$Si signatures of diatoms within the sediments are generally unaffected by diagenetic alteration (e.g. Egan et al., 2012; Wetzel et al., 2014; Ehlert et al., 2016).

On line 91, the authors should be more specific than "high amounts" – are you referring to high concentrations, fluxes, or both?

We are referring to concentrations here, accordingly concentration values for Si(OH)$_4$ and NO$_3$- are now given in brackets. Line 108-109: 'Along the Peruvian margin the main source for the high amounts of upwelled nutrients (30 µmol L$^{-1}$ for both Si(OH)$_4$ and NO$_3^-$; Bruland et al., 2005)'

What do the +/- signs on lines 97 onwards represent?

For the $\delta^{30}$Si the +/- always indicates the 2SD external reproducibility, only for the $\delta^{15}$NO3- values taken from Rafter 2012 the +/- indicates the 1SD variability of several water masses. This has now been added to the text in brackets in lines 114 and 115.

One line 114, the authors could add a few words to explain why the steady-state system is appropriate here. This arises again mater in the manuscript, but I think it would help to clarify the choice here as well.

Based on observations of Ehlert et al. 2012, the corresponding reference was added to Lines 132-33.

On line 138, the authors should remove "in study of".

Line 164: 'study of' has been removed accordingly.

The sentence on line 178 is not complete – please rewrite. Also the short paragraph on line 186 onwards seems a little misplaced – I'd suggest the end of that section is rephrased.

The paragraph (Line 379-386) was rephrased as follows:

'This scenario is supported by a marked reduction in the concentrations of sedimentary redox sensitive trace metals such as molybdenum and rhenium (Salvatteci et al., 2014b; Sifeddine et al., 2008). However, these conditions were not constant, instead short-term variations during both the LIA and the CWP are reflected for example mirrored by changes in diatom abundances, productivity sensitive element ratios (Br/Fe) and $\delta^{15}N_{bulk}$ values (Fleury et al., 2015). These proxy records indicate multidecadal shifts between arid/humid conditions during the CWP and particularly the LIA during which marked short-term periods of arid conditions occurred (Fleury et al., 2015) (Fig. 3).'

Line 290: please avoid using "a bit lower" – rephrase.

The sentence was rephrased accordingly:

Line 446: 'The shift towards a higher 1:2 $NO_3^-$:$Si(OH)_4$ utilization during both the CWP and LIA (arid) indicates enhanced utilization of $Si(OH)_4$ over $NO_3^-$ leading to $Si(OH)_4$ limitation as indicated by high $Si(OH)_4$ utilization rates between 40% and 90%, and lower $NO_3^-$ utilization rates between 25% and to 80% (Fig. 6b).'

Line 327: I'm not sure what you mean by "horizontal alignment" – could you please clarify?

The text (Line 489-490) was changed accordingly: 'horizontal alignment of the $\delta^{15}N_{bulk}$ versus $\delta^{30}Si_{BSi}$ values'

Line 311: Is there no means of assessing changes in downcore phytoplankton assemblages, as a comparison to the modern data from Sanchez et al? Biomarkers?

It is possible to compare the diatom assemblages as done by Fleury et al., 2015. What we wanted to highlight here is that there are no modern $\delta^{15}N$ and $\delta^{30}Si$ isotope values available to compare modern El-Nino conditions with El-Nino-like conditions in the past as claimed for the LIA.

Figure caption 4: The caption points towards figures c-e, when they should be figures b-d.

The figure caption has been corrected accordingly.

Figure 6: The fonts are too small in places.

The fonts of the figure have been enlarged accordingly (see figure attached).

Additional references:

Demarest, M.S., Brzezinski, M.A. and Beucher, C., 2009. Fractionation of silicon isotopes during biogenic silica dissolution. Geochimica et Cosmochimica Acta, 73: 5572-5583.

Egan, K. et al., 2012. Diatom silicon isotopes as a proxy for silicic acid utilisation: A Southern Ocean core top calibration. Geochimica et Cosmochimica Acta, 96: 174-192.

Sutton, J., Varela, D., Brzezinski, M.A. and Beucher, C., 2013. Species dependent silicon isotope fractionation by marine diatoms. Geochimica et Cosmochimica Acta, 104: 300-309.

Wetzel, F., de Souza, G. and Reynolds, B., 2014. What controls silicon isotope fractionation during dissolution of diatom opal? Geochimica et Cosmochimica Acta, 131: 128-137.

The references have been included in the text and accordingly been added to the reference list.

First, I think the $d^{30}Si$ and $d^{15}N$ relationship in Figure 4A must be examined in a more robust manner. For example, I don't think it is appropriate to simply state that the surface sediment core measurements "remain close to the respective 1:1 utilization line" for nitrate to silicate in Figure 4A. This needs to be shown, regardless of this approach being previously published by Ehlert et al. (2015). Specifically, the study would be improved with a more precise quantification of this relationship in Figure 4A. Similarly, the frequent use of "correlation" in the text (n=9) is not supported with any statistics. This must be fixed.

We added a more comprehensive explanation of what is shown in Fig. 4a and how it relates to our results in S1 and Fig 4 b-d. To address this issue we improved the text explaining Fig. 4 a and the conclusions we draw from it. This point will also be further addressed below, under the last point concerning the restructuring of the Results and Discussion sections.

This point will be also further addressed below, under the last point concerning the restructuring of the Results and discussion.

The improper use of the word "correlation" was checked throughout the whole manuscript. See lines: Lines 25, 99, 572, 574 and 575, as well as in the figure captions.

Second, I think it would improve this study if the "nutrient utilization" plots in Figures 2 and 3 were separated instead of overtop each other. The estimated nitrate and silicate utilization by phytoplankton should be on separate plots.

Following the reviewer's suggestion we changed figure 2 to improve the visibility of the nutrient utilization. However we did not separate $NO_3^-$ and $Si(OH)_4$ utilization given that the purpose of this figure is exactly to illustrate the relationship of the changes in both systems. See revised Figures 2 below.

Another strong recommendation is to compile the sediment core measurements per latitude in Figure 3 in a more comprehensible way. The reader will have a much easier time understanding the relationships in Figure 3 if they: (1) Show one measurement at a time and (2) Are "stacked" from low to high southern latitudes. In this new figure, the bulk sediment d15N for all sites will be shown 'stacked' in one column that moves from 11ºS at the top to 15ºS at the bottom. This new arrangement of the Figure 3 data will include the same data, but in a more easily understood arrangement and will allow the reader to identify the spatial and temporal variability of each proxy.

Figure 3 was modified and we now separated the different proxies (see below). Furthermore, the records are now sorted from 11 to 15ºS from top to bottom.

Fourth, the use of sedimentary percentages is not appropriate; we must see the percentage data expressed as a mass accumulation rate or MAR. This needs to be changed before I would have confidence in the interpretation (any interpretation) of the percentage measurements. The addition of MARs may actually improve the interpretation of the data, since there seems to be some confusion in the interpretation of these measurements (see Lines 204 and on).

For presentation of the BSi and TN data in Figures 2 and 5, we now calculated the respective accumulation rates based on data presented by Gutierrez et al., 2009 given for 12ºS and 14ºS, respectively, as there are no accumulation rates available for cores M77/2-24, 005 and 003. As the BSi (wt%) data is new we still show these values in Figure 3.

However, we want to point out that there is no confusion about the interpretation of BSi data. As visible in the comparison of the records of core M77/2-24 with the previously published core M77/1-470 from the same latitude the BSi (wt%) is overlapping during the CWP, but BSi values of core 24 remain high throughout the LIA and $\delta^{30}Si$ signatures of core 24 are higher by about 0.5‰ at all times. From line 204 onwards it is discussed how this difference in $\delta^{30}Si$ can occur in the 2 records located so close to each other.

Text added:

Lines 169-174: Unfortunately, no material was left of the cores studied to estimate dry bulk densities to calculate mass accumulation rates (MAR), therefore values were used from cores BO413 (12 ºS) and BO406 (14 ºS; Gutierréz et al. 2009) which were generally close to 0.02 (g cm$^{-2}$ yr$^{-1}$) during the LIA and 0.03 (g cm$^{-2}$ yr$^{-1}$) during the CWP. The exact bulk MAR values (g cm$^{-2}$ yr$^{-1}$) for each time period were multiplied by the fractional concentration of BSi and TN (Fleury et al., 2015) to calculate the MAR BSi and MAR TN (Figs. 2 and 5).

Finally, I don't think the Discussion section is the location for describing every individual wiggle of the observations and /or the estimated source nitrate d15N. I think that an improved manuscript would have a robust statistical examination of the surface sediment d15N and d30Si (from Figure 4) in the Results section followed by text that describes the temporal variability of Figures 4 and 5. In this way, the Discussion section can be used to discuss the observed / estimated spatial and temporal changes, which will be more easily understood and (I think) enjoyable for the reader. No one wants to read a listing of which way the wiggles are wiggling and when.

Following the suggestion of the reviewer the Results and Discussion section was restructured i.e. split. The Equation for Nutrient utilization was moved to the Methods section (new section 2.4; Lines 208-227).

To address the statistical examination we added a supplementary figure (see below) that gives all linear regressions and r$^2$ used for the calculations of the $\delta^{15}NO_3^-$ source signatures. For the surface sediment samples a statistical examination is of little gain due to the small sample amount concerned, however we added a more detailed explanation about the information gained by Fig. 4a that should resolve this issue.

Lines 238-264:' Based on modern observations from the water column it is known that $NO_3^-$ and $Si(OH)_4$ are incorporated in a 1:1 ratio when diatoms dominate the phytoplankton assemblage (Brzezinski, 1985; Ragueneau et al., 2000) this ratio, however, can vary between 2:1 and 1:2 on the shallow Peruvian shelf (Grasse et al., 2016). Assuming this is true for the past as well, a change in relative uptake will be visible in the $d^{15}N_{Bulk}$ versus $d^{30}Si_{BSi}$ relationship in surface sediments. We therefore applied a direct comparison of both isotope signatures in the surface sediments, following an approach of Ehlert et al. (2015). To evaluate how modern sediments record the relative nutrient utilization of both $NO_3^-$ and $Si(OH)_4$, surface sediment values were sorted by latitude (9, 11, 12 and 15ºS) and compared to their respective $d^{15}NO_3^-$ and $d^{30}Si(OH)_4$ source and utilization relationship at each location.

First, based on the known $d^{30}Si(OH)_4$ and $NO_3^-$ source signatures of subsurface waters we can calculate a theoretical utilization assuming 1:1 incorporation ratios of both $Si(OH)_4$ and $NO_3^-$. There are several factors that may cause $d^{15}N$:$d^{30}Si$ ratios to deviate from this 1:1 utilization uptake ratio of both nutrients: 1) under iron-limitation more $Si(OH)_4$ relative to $NO_3^-$ will be taken up resulting in heavier silicified diatoms and shifting to a 1:4 N:Si uptake ratio (Hutchins and Bruland, 1998; Franck et al., 2000); 2) under prevalence of non-siliceous phytoplankton groups more $NO_3^-$ than $Si(OH)_4$ is incorporated (Conley and Malone, 1992; Wilkerson and Dugdale, 1996), the ratio might shift up to 15:1 (Grasse et al., 2016). Therefore, the deviation from the 1:1 $NO_3^-$ to $Si(OH)_4$ ratio can serve as an indicator for the degree of relative utilization of $NO_3^-$ over $Si(OH)4$ (Grasse et al., 2016; see utilization schematic indicated in Figure 4a).

The surface samples from 9ºS (Fig. 4) indicate low to moderate utilization of both nutrients. The sample values, however, plot beneath the 1:1 utilization line, reflecting higher $NO_3^-$ than $Si(OH)_4$ utilization, which is supported by the calculated $NO_3^-$ and $Si(OH)_4$ utilization as based on a steady state calculation (Fig. 2b). The data from 11ºS (Fig. 4a) all plot close to the middle of the respective 1:1 utilization line at about the middle indicating about 50% utilization for both $NO_3^-$ and $Si(OH)4$, agreeing with the calculated utilization values shown in Fig. 2b. For 15º S the sample values plot further to the left of the 1:1 utilization, indicating higher utilization of $Si(OH)4$ than $NO_3^-$ (Fig. 4a, 2b). This displacement to the left supports a $NO_3^-$:$Si(OH)_4$ ratio of 1:2 or 1:4, which agrees with direct observations from the surface waters in areas of strong upwelling (Grasse et al., 2016).'

Further, the calculation of the $\delta^{15}NO_3^-$ source based on linear regression was also moved to the Methods (now section 2.4; Lines 266-274). This paragraph was also rephrased accordingly:

Lines 257-265:' For the past time periods of the CWP and LIA conditions are thought to shift between two states (1) high upwelling intensity, high productivity and strong N-loss processes and (2) low productivity and N-loss intensity. For either state we assume that   relationship between $\delta^{30}Si_{BSi}$ and $\delta^{15}N_{bulk}$ signatures should thus also be a direct indicator of the $\delta^{15}NO_3^-$ source signatures. . Depending on the exact conditions the uptake of $NO_3^-$ to $Si(OH)_4$ might be in a ratio of for example 1:1, 2:1 or 1:2. If the conditions remain stable over the time periods samples should represent a linear relationship similar to the utilization lines indicated in Fig. 4. Assuming such a linear relationship of $\delta^{15}N_{bulk}$ to $\delta^{30}Si_{BSi}$ samples for past periods, it is possible to first calculate the respective $\delta^{15}NO_3^-$ source values and then further quantify the amount of $NO_3^-$ utilization.'

Lines 291-294:'For all time periods and latitudes, the linear regressions as well as correlation coefficient $(r^2)$ are given in the supplements (Fig. S1). The results are presented in the following as the resulting $\delta^{15}NO_3^-$ source values and the theoretical ratio of nutrient utilization (i.e. 1:1 or 2:1, 15:1, etc.; Fig. 4. b-d) for each latitudinal range to compare the latitudinal trends between the CWP and LIA'

Additionally, paragraphs within the previous Results and Discussion section mainly describing results were taken omitted (see Lines 424-433; 458-466; 522-535) or rephrased (Line 467-471) and a separate Results section was written (subsections 3.1 Biogenic opal and silicon isotope signatures and 3.2 d$^{15}$NO$_3^-$ source signatures, nutrient utilization and supply) see lines 336- 362.

Line 29: I don't think "humid conditions" is appropriate at this point; it is not common knowledge, it is not supported with evidence, and is out of place in the Abstract.
'humid conditions' has been excluded from this sentence as suggested by the reviewer.

The isotope effects of nitrate assimilation were estimated at sites across the Pacific basin in Rafter and Sigman 2016. This is blatant self-promotion, but it is entirely relevant when citing reasonable isotope effects in the tropical Pacific (we found an average of 6 per mil). We also identify the origin of variability between Rayleigh (closed system) versus Open System isotopic fractionation.
The given fractionation factors actually match values calculated for the Peruvian shelf, the appropriate references were added accordingly.
Line 45: .... , which agree well with estimates for the Peruvian shelf (Ehlert et al., 2012; Mollier-Vogel et al., 2012; Grasse et al., 2016).
However, the reference to Rafter and Sigman 2016 was added in Line 116, as it also gives a detailed overview about the d$^{15}$NO$_3^-$ values within the EUC.

Line 81: remove "been"
Line 98 'been' has been removed

89: remove "too"
Line 106 (previously 89) 'to' has been removed accordingly

99: When dealing with nitrate d15N, a difference in source waters of 5.5 to 7.0 per mil is quite large. Especially considering the small spatial difference between these sources.
Line 117: As pointed out by the reviewer a 1.5 per mil difference is not 'slight', therefore we removed 'slightly' from the sentence and replaced it with 'about 1.5‰'.

201: Sedimentary concentration measurements are common, but they are not useful or appropriate proxies in 2018. MASS ACCUMULATION RATES are necessary. Apply and then we can re-examine the records.

As now stated in line 169 and also line 303, there are unfortunately no accumulation rates available for the cores, as no dry bulk density was analyzed to calculate MAR for cores M772-024TC, 005TC and 003TC. Therefore, we used accumulation rates reported of cores BO406-13 and BO406-5 reported for the last 600 years by Guiterréz et al. 2009 to estimate the MAR TN and MAR BSi values in our cores. These data is shown in Figures 2 and 5 now. We left the actual BSi values in Figure 3, also as a comparison of the values with core M77/1-470. As accumulation rates mainly decrease during the LIA in comparison to the CWP, there should be little difference in the comparison of BSI and AR BSi concerning these two cores.

See also Line 169- 174:' Unfortunately, no material was left of the cores studied to estimate dry bulk densities to calculate mass accumulation rates (MAR), therefore values were used from cores BO413 (12 ºS) and BO406 (14 ºS; Gutierrez et al. 2009) which were generally close to 0.02 (g cm$^{-2}$ yr$^{-1}$) during the LIA and 0.03 (g cm$^{-2}$ yr$^{-1}$) during the CWP. The exact bulk MAR values (g cm$^{-2}$ yr$^{-1}$) for each time period were multiplied by the fractional concentration of BSi and TN (Fleury et al., 2015) to calculate the MAR BSi and MAR TN (Figs. 2 and 5).'

List of relevant changes:

- the descriptions of the results have been improved
- figure 3 has been corrected and been improved based on the suggestions of the reviewers
- information has been added about the influences of dissolution and species-specific fractionation for the fractionation factor of Si
- A comparison of the d30Si values with the diatom assemblages has been added
- the sampling resolution has been clarified in the methods section
- the explanation of figure 4 a has been improved and a plot showing all linear regressions was added in S1
- the way how the nutrient utilization is plotted has been changed in figures 2 and 5
- MAR values have been calculated based on DBD values from nearby cores; MAR values are shown now in figures 2 and 5
- The results and discussion sections have been split and text passages including equations have been moved to the methods section.

[revised manuscript text omitted]

In contrast, the $\delta^{30}Si_{BSi}$ signatures north of 10°S are highly variable (0.3-0.9‰) reflecting an overall lower degree of utilization (20-40%; Doering et al., 2016; Ehlert et al., 2012) (Please note that all Si and N utilization calculations are based on an open steady-state system, which was shown to more accurately reflect the natural system off Peru, see Ehlert et al., (2012). At the central Peruvian shelf (10-12°S), where subsurface $[O_2]$ is <20 µmol $L^{-1}$ (Fig. 2a), the subsurface source value of $\delta^{15}NO_3^-$ increases to 8.6‰ due to denitrification (Mollier-Vogel et al., 2012). The $\delta^{30}Si_{BSi}$ and $\delta^{15}N_{bulk}$ values both increase to 0.9 ± 0.1‰ and 6.1 ± 0.8‰, respectively, reflecting higher $Si(OH)_4$ utilization (20-60%) but a decrease inlower $NO_3^-$ utilization compared to the northern part of the study area (15-65%, Fig. 2b) associated with increased upwelling intensity, high nutrient re-supply and higher consumption via diatom productivity. In the southernmost part of the shelf (13-16°S) highest productivity and upwelling intensity prevail, leading to a further increase in the subsurface $\delta^{15}NO_3^-$ signature of up to 12.5‰ at 15°S (southernmost surface sediment station of the study area), whereas surface sediment mean $\delta^{30}Si_{BSi}$ and $\delta^{15}N_{bulk}$ values further increase to 1.1 ± 0.1‰ and 8.9 ± 1.4‰ reflecting moderate $Si(OH)_4$ utilization of 50-70% ($\delta^{30}Si_{BSi}$) and $NO_3^-$ utilization of 20-70% ($\delta^{15}N_{bulk}$). The supply for 
[revised manuscript text omitted]
 this is true for the past as well, a change in relative uptake will be visible in the $\delta^{15}N_{Bulk}$

versus $\delta^{30}Si_{BSi}$ relationship in surface sediments. We therefore applied a direct comparison of both isotope signatures in the surface sediments, following an approach of Ehlert et al. (2015). To evaluate how modern sediments record the relative nutrient utilization of both $NO_3^-$ and $Si(OH)_4$, surface sediment values were sorted by latitude (9, 11, 12 and 15°S) and compared to their respective $\delta^{15}NO_3^-$ and

$\delta^{30}Si(OH)_4$ source and utilization relationship at each location.

  First, based on the known $\delta^{30}Si(OH)_4$ and $NO_3^-$ source signatures of subsurface waters we can calculate a theoretical utilization assuming 1:1 incorporation ratios of both $Si(OH)_4$ and $NO_3^-$. There are several factors that may cause $\delta^{15}N$: $\delta^{30}Si$ ratios to deviate from this 1:1 utilization uptake ratio of both nutrients: 1) under iron-limitation more $Si(OH)_4$ relative to $NO_3^-$ will be taken up resulting in heavier silicified diatoms and shifting to a 1:4 N:Si uptake ratio (Hutchins and Bruland, 1998; Franck et al., 2000); 2) under prevalence of non-siliceous phytoplankton groups more $NO_3^-$ than $Si(OH)_4$ is incorporated (Conley and

Malone, 1992; Wilkerson and Dugdale, 1996), the ratio might shift up to 15:1 (Grasse et al., 2016).

Therefore, the deviation from the 1:1 $NO_3^-$ to $Si(OH)_4$ ratio can serve as an indicator for the degree of relative utilization of $NO_3^-$ over $Si(OH)_4$ (Grasse et al., 2016; see utilization schematic indicated in Figure 4a).

  The surface samples from 9°S (Fig. 4) indicate low to moderate utilization of both nutrients.

The sample values, however, plot beneath the 1:1 utilization line, reflecting higher $NO_3^-$ than $Si(OH)_4$

utilization, which is supported by the calculated $NO_3^-$ and $Si(OH)_4$ utilization as based on a steady state calculation (Fig. 2b). The data from 11°S (Fig. 4a) all plot close to the middle of the respective 1:1

utilization line at about the middle indicating about 50% utilization for both $NO_3^-$ and $Si(OH)_4$, agreeing with the calculated utilization values shown in Fig. 2b. For 15° S the sample values plot further to the left of the 1:1 utilization, indicating higher utilization of $Si(OH)_4$ than $NO_3^-$ (Fig. 4a, 2b). This displacement to the left supports a $NO_3^-$:$Si(OH)_4$ ratio of 1:2 or 1:4, which agrees with direct observations from the surface waters in areas of strong upwelling (Grasse et al., 2016).

For the past time periods of the CWP and LIA conditions are thought to shift between two states (1) high upwelling intensity, high productivity and strong N-loss processes and (2) low productivity and N-loss intensity. For either state we assume that   relationship between $\delta^{30}Si_{BSi}$ and $\delta^{15}N_{bulk}$ signatures should thus also be a direct indicator of the $\delta^{15}NO_3^-$ source signatures.

. Depending on the exact conditions the uptake of $NO_3^-$ to $Si(OH)_4$ might be in a ratio of for example 1:1, 2:1 or 1:2. If the conditions remain stable over the time periods samples should represent a linear relationship similar to the utilization lines indicated in Fig. 4. Assuming such a linear relationship of $\delta^{15}N_{bulk}$ to $\delta^{30}Si_{BSi}$ samples for past periods, it is possible to first calculate the respective $\delta^{15}NO_3^-$ source values and then further quantify the amount of $NO_3^-$ utilization.

In order to estimate past changes in the $\delta^{15}NO_3^-$ source values the $\delta^{30}Si_{BSi}$ and $\delta^{15}N_{bulk}$ values for both time periods, CWP and arid and humid periods during the LIA, were separately plotted against each other (Fig. 4c-e) and $\delta^{15}NO_3^-$ was calculated based on the linear function assuming that the source

$\delta^{30}Si(OH)_4$ signature always remained stable at 1.5‰ over time:

[revised manuscript text omitted]

2TC) presented in this study were on average 0.8‰ lower during the LIA than during the CWP (Fleury et al., 2015; Fig. 3 h,i, k). The $\delta^{15}$N$_{bulk}$ values reported for core 005-3TC (12°S) are close to values of nearby core B0406-13 (Gutiérrez et al., 2009; Fig. 3i). Similarly, the $\delta^{15}$N$_{bulk}$ values of core 003-2TC

(15°S) agree well with previously published $\delta^{15}$N$_{bulk}$ record of core B0405-6 (14°S, Fig. 3 j, k; Gutiérrez et al., 2009); Overall, productivity and nutrient utilization proxies (BSi and $\delta^{30}$Si$_{BSi}$) varied in phase with

$\delta^{15}$N$_{bulk}$ signatures at all core locations (Fig. 3).

**3.2 $\delta^{15}NO_3^-$ source signatures, nutrient utilization and supply**

~~The isotopic compositions of samples from all cores generally plot close to the 1:1 utilization lines indicating a $\delta^{15}NO_3^-$ source value of 8 to 9‰, similar to present day subsurface waters between 10°S and 15°S (Fig. 4a, b). Furthermore, the $\delta^{15}N_{bulk}$ values and calculated subsurface $\delta^{15}NO_3^-$ source values indicate a similar southward increase from 7‰ to 10‰ between 11°S and 15°S during the CWP (Fig. 4b).~~

During the CWP the calculated $\delta^{15}NO_3^-$ source signatures based on eq. (2) and (3) result in values of 7.6‰ at 11ºS, 8.6‰ at 12ºS and 10.4‰ between 14ºS and 15ºS during the CWP (Fig. 4b; S1 a-c), which reflects a southward increase in $\delta^{15}NO_3^-$ source signatures as observed today (Fig. 2a). Based on these $\delta^{15}NO_3^-$ values the nutrient utilizations estimated based on eq. (1) range between 30-90% for $NO_3^-$ and 40-100% for $Si(OH)_4$ (Fig. 5a). During the LIA (arid) similar values are calculated $d^{15}N$ 8‰ between 11ºS and 12ºS and 9‰ between 14ºS and 15ºS. The respective nutrient utilization ranges between 2 to 70% for $NO_3^-$ and 20 to 85% for $Si(OH)_4$. During the humid phases of the LIA the  calculated $\delta^{15}NO_3^-$ source values were lower reaching  6‰  between 11°S and 12°S and 7.5‰  between 14°S and 15°S (Fig. 4d).  The calculated $NO_3^-$ utilization  ranges between 70 and 90% and  The  $Si(OH)_4$ utilization  ranges between 6 and 60% (Fig. 5c).

 The mass accumulation rates of TN (MAR TN) were lower by around 0.02 (g/cm$^2$/yr)  lower during the CWP than today and lowest (<0.02 g/cm$^2$/yr) during the LIA, with little difference between humid and arid phases. The MAR BSi values  were generally higher by about  0.1 - 0.35 (g/cm$^2$/yr) during the CWP and LIA (arid) than today with highest values at 15°S, and more similar values to today during the LIA (humid) (Figs. 2, 5).

**4. Discussion**

The climate of the last 600 years is divided into two climatic phases consisting of the CWP (since 1800 AD) and the LIA (ca. 1400 to 1800 AD). Off Peru, the CWP has been characterized by dry (arid) conditions, high upwelling intensity, as well as high productivity and strong N-loss processes, reflecting overall dominant La-Niña conditions (Fleury et al., 2015; Salvatteci et al., 2014b; Sifeddine et al., 2008). In contrast, the LIA has been shown to be characterized by lower productivity and low denitrification intensity for the present day main upwelling area between 10°S and 15°S (Díaz-Ochoa et al., 2009; Salvatteci et al., 2014b; Sifeddine et al., 2008). Such conditions are more similar to modern El-Niño events in the area and thus generally referred to as El-Niño-like conditions (Clement et al., 2000). Previous paleo-reconstructions agreed that the latter conditions were a consequence of larger scale climatic changes induced by weakening of the Walker circulation and reduction of the South Pacific subtropical High (SPSH), as well as by a southward shift of the mean position of the Intertropical Convergence Zone (ITCZ) and the associated precipitation belt (Fleury et al., 2015; Sachs et al., 2009; Salvatteci et al., 2014b; Sifeddine et al., 2008). This was inferred to result in a deepening of the nutricline and reduced surface productivity and more oxygenated subsurface waters, as indicated by lower BSi and TOC concentrations (*this study*; (Ehlert et al., 2015; Gutiérrez et al., 2009;

Salvatteci et al., 2014a) and Si/Fe ratios (Fleury et al., 2015).  This scenario is supported by a marked reduction in the concentrations of sedimentary redox sensitive trace metals such as molybdenum and rhenium (Salvatteci et al., 2014b; Sifeddine et al., 2008). However, these conditions were not constant  instead short-term variations during both  the LIA and the CWP

for example mirrored by changes in diatom abundances, productivity sensitive element ratios (Br/Fe) and $\delta^{15}N_{bulk}$ values (Fleury et al., 2015). These proxy records  indicate multidecadal shifts between arid/humid conditions during the CWP and particularly the LIA during which marked short-term periods of arid conditions occurred (Fleury et al., 2015) (Fig. 3).

The well-studied biogeochemical evolution of the Peruvian shelf over the last 600 years and the significant differences in productivity and subsurface oxygenation between the CWP and the LIA are the basis for our study to gain new insights into the relationship between nutrient utilization and denitrification via $\delta^{30}Si_{BSi}$ and $\delta^{15}N_{bulk}$ records.

**4.1 Changes in BSi production and $\delta^{30}Si_{BSi}$ during the last 600 years**

The BSi concentrations and $\delta^{30}Si_{BSi}$ signatures at 12°S (005-3TC) and 15°S (003-2TC)

are lowest during the LIA (Fig. 3c,e), in agreement with previously published records from 11°S (M77/1-

470; Fig. 3a) and 14°S (Ehlert et al., 2015; Fig. 3d). An exception is core 024-5TC (Fig. 3a) from 11°S, where $\delta^{30}Si$ mean values of the LIA (1.3 ± 0.4‰) are similar to CWP mean values (1.4 ± 0.1‰).

Furthermore, both the BSi concentrations and $\delta^{30}Si_{BSi}$ of core 024-5TC are significantly higher during the LIA than at nearby core M77/1-470 (Fig. 2a; Ehlert et al. 2015). Such consistently high BSi and $\delta^{30}Si$

values reflect a high degree of Si(OH)$_4$ utilization–generally associated with strong upwelling conditions  at site 024-5TC during the LIA

. This is, however, not supported by other productivity proxies such as the Si/Fe ratio and the total diatom abundance for core 024-5TC, which do not indicate similarly high

BSi production during the LIA and the CWP (Fleury et al., 2015). However, comparison with the cumulative diatom assemblage indicates overall little difference in the amount of upwelling and coastal planktonic diatom species between the LIA and the CWP at 11°S (Fig. 3f), with intervals of reduced abundances of upwelling species of generally less than 50 years, much shorter than the 100 to 150 years intervals observed at 12 and 15°S. Furthermore, the finely laminated sediments do indicate short periods of higher productivity during the LIA in phase with more arid conditions (Fig. 3, grey shadings; for details see (Fleury et al., 2015). Accordingly, the high mean BSi and $\delta^{30}Si_{BSi}$ values obtained from core

024-5TC may be an artifact of the low sampling resolution with only two $\delta^{30}Si_{BSi}$ samples between 1700

and 1800 AD , as $\delta^{30}Si_{BSi}$ analysis do not cover all short events (~50 years) of
reductions in the abundance of upwelling diatom species (Fig. 3f). Alternatively, the increase in $Si(OH)_4$
utilization decoupled from an increase in diatom abundance (Fleury et al., 2015) not shown here) may
indicate stronger silicification of the diatom frustules, as often observed under iron (Fe)-deficient
conditions further associated with an increase in the $Si(OH)_4:NO_3^-$ incorporated by the diatoms (De La
Rocha et al., 2000; Takeda, 1998; Wilken et al., 2011).

**4.2 Impact of denitrification versus nutrient utilization on the records**

**4.2.1 Disentangling nutrient utilization and N-loss processes: Changes in the source water nitrate isotopic composition**

Given the significant changes in upwelling intensity, productivity and subsurface oxygenation (and thus
N-loss) between the LIA and CWP (Fig. 3), we first investigated if and how the preformed $\delta^{15}NO_3^-$ signal
changed with latitude (between 11°S and 15°S), and then used this information to reconstruct how
nutrient supply and utilization changed during these time periods.
is in agreement
surface sediment $\delta^{30}Si_{BSi}$ versus $\delta^{15}N_{bulk}$   the southward increase in
source $\delta^{15}NO_3^-$ signatures (Fig. 4a; stars), but remain close to the respective 1:1 utilization line for $NO_3^-$
$:Si(OH)_4$.
Our results show that the correlation of $\delta^{30}Si_{BSi}$ to $\delta^{15}N_{bulk}$ during the CWP is comparable to
modern values between 10°S and 15°S (Figs. 4a,b). The isotopic compositions of samples from all cores
generally plot close to the 1:1 utilization lines indicating a $\delta^{15}NO_3^-$ source value of 8 to 9‰, similar to
present day subsurface waters between 10°S and 15°S (Fig. 4a, b). Furthermore, the $\delta^{15}N_{bulk}$ values and
calculated subsurface $\delta^{15}NO_3^-$ source values indicate a similar southward increase from 7‰ to 10‰
between 11°S and 15°S during the CWP (Fig. 4b). Thus, we suggest that the net increase in $\delta^{15}N_{bulk}$ from
North to South during the CWP resembles the increase observed in modern surface sediments (Figs. 2
and 5a). However, unlike todays surface sediment data, the cores at 11°S and 12°S show substantially
higher $\delta^{30}Si_{BSi}$ values during the CWP. According to the steady state calculations (3) this reflects high
$Si(OH)_4$ utilization of up to 100%, whereas the $NO_3^-$ utilization only reaches 80% at maximum (Fig. 5a).
Based on the classification of Fleury et al. 2015, we further distinguished between samples from
humid periods (El Niño-like conditions) and arid periods (La Niña-like conditions) for the LIA (Figs. 4
and 5). This differentiation clearly highlights that samples from LIA (arid) phases are similar to samples
from the CWP with $\delta^{15}NO_3^-$ source signatures of 8‰ to 9‰ between 11°S and 15° S, respectively. The
shift towards a higher 1:2 $NO_3^-:Si(OH)_4$ utilization during both the CWP and LIA (arid) indicates enhanced utilization of Si(OH)$_4$ over NO$_3^-$ leading to Si(OH)$_4$ limitation as indicated by high Si(OH)$_4$
utilization rates between 40% and 90% and lower NO$_3^-$ utilization rates
 25%   80% (Fig. 5b). Such a decoupling of Si and N within diatoms can be caused by
biogeochemical changes, such as iron availability altering the Si:N uptake dynamics (Hutchins and
Bruland, 1998; Takeda, 1998) whereby elevated Si:N ratios are characteristic for Fe-limited diatom
communities (Takeda, 1998). Accordingly, increased uptake of Si over N can lead to a Si(OH)$_4$ limitation
as found during the CWP and LIA arid phases at 11°S to 12°S. The reason may have been that less Fe
was upwelled at the narrow shelf between 11°S and 16°S, which led to Fe-limitation during progressing
diatom blooms (Doering et al., 2016).

During the humid phases of the LIA the $\delta^{15}$NO$_3^-$ source signatures decrease to 6‰ (11-12 °S)
and 7.5‰ (14-15 °S) further indicating increased NO$_3^-$ over Si(OH)$_4$ utilization (ratios of 2:1 and 15:1;
Fig. 4 d) as supported by increased NO$_3^-$ utilization and decreased Si(OH)$_4$ utilization (Fig. 5c). While
$\delta^{30}$Si$_{BSi}$ values remain remarkably stable, $\delta^{15}$N$_{bulk}$ values show a wide range between 4.8‰ and 6.7‰
potentially reflecting enhanced NO$_3^-$ limitation prevailing during these humid phases. Such a  shift
towards increasing $\delta^{15}$N$_{bulk}$ values with consistently low $\delta^{30}$Si$_{BSi}$ values can be caused by weaker
denitrification due to the higher subsurface oxygenation (only suboxic and not anoxic conditions)
reconstructed for the area together with decreased upwelling conditions at the time (Salvatteci et al.,
2014b; Sifeddine et al., 2008). Unfortunately, to date there are no isotopic measurements available for
either present day El-Niño $\delta^{15}$NO$_3^-$ or $\delta^{30}$Si(OH)$_4$ signatures of Peruvian water masses in order to better
evaluate the effects of these significant changes in climatic forcing. Overall, our results indicate much
higher NO$_3^-$ utilization over Si(OH)$_4$ utilization (Fig. 5c), which is in agreement with phytoplankton
assemblage analyses during El-Niño events indicating dominating productivity of non-siliceous
phytoplankton groups (Sanchez et al., 2000). Accordingly, primary productivity dominated by non-
siliceous phytoplankton leads to enhanced NO$_3^-$ over Si(OH)$_4$ uptake. Although similar conditions are
found offshore the Peruvian shelf today, these surface waters originate from the shelf area where diatom
blooms prevail, thus being already depleted in Si(OH)$_4$ and not providing an adequate analogue for the
conditions observed during the LIA.

However, while NO$_3^-$ limitation seems to have prevailed at all latitudes during the LIA (humid)
(i.e. all sample fall between the 2:1 and 15:1 utilization lines), only the data at 11-12°S indicate a $\delta^{15}$NO$_3^-$
source value close to 6‰. In contrast samples from 14-15°S show  higher $\delta^{15}$N$_{bulk}$ values up to
7.3 ‰, potentially indicating a slight increase in $\delta^{15}$NO$_3^-$ to up to 7.5‰ due to ongoing but decreased denitrification at 14 to 15°S during this time. It should be noted that it was not possible to calculate $\delta^{15}NO_3^-$ values based on the linear function (eq. (2)) during this period, due to near horizontal alignment of the $\delta^{15}N_{bulk}$ versus $\delta^{30}Si_{BSi}$ values. Alternatively, we cannot exclude that under El-Niño-like conditions, the hydrodynamic conditions off Peru may be better described by a Rayleigh-type model (closed system) instead of the steady state model, which was reported to better describe the modern conditions (Ehlert et al., 2012). Applying a Rayleigh-type model calculation, $\delta^{15}N_{bulk}$ values of 7.3‰ during the LIA (humid) would reflect approximately 75% utilization assuming the $\delta^{15}NO_3^-$ source value remaining at 6‰ and an enrichment factor of -5‰. As upwelling of nutrient-rich water masses was diminished during this time, a closed system scenario is possibly to be more appropriate to interpret the isotopic signatures and would be in agreement with a more oxygenated (suboxic) water-column at the time reducing subsurface denitrification (Briceño-Zuluaga et al., 2016; Gutiérrez et al., 2009; Salvatteci et al., 2014b; Sifeddine et al., 2008). However, due to the lack of modern data for comparison with El-Niño events for comparison we assume a decrease to a $\delta^{15}NO_3^-$ source value of 6‰ between 11°S and 12°S and 7.5‰ between 14°S and 15°S during the LIA (humid) for the following discussion, still indicating a slight southward increase given that denitrification was not shut down completely (suboxic conditions as indicated by trace metal studies; Salvatteci et al., 2014b; Sifeddine et al., 2008) .

**4.2.2 Latitudinal variation The effects of changes in the nutrient supply during the CWP and LIA**

Comparison of $\delta^{30}Si_{BSi}$ and $\delta^{15}N_{bulk}$ values for the past ~600 years indicates that the $\delta^{15}NO_3^-$ source values between 11°S and 15°S during the CWP and LIA (arid) were comparable to today yielding southward increasing values of 7‰ to 10‰ caused by subsurface denitrification under oxygen-depleted conditions (Fig. 2; Fig. 5a, b). For both the CWP and LIA (arid) calculated nutrient utilization indicates higher $Si(OH)_4$ utilization (ranging between 30 and 100%) over $NO_3^-$ utilization (ranging between 20 and 90%), similar to modern utilization at 10-15°S (Figs. 2, 5). However, These estimates indicate higher $Si(OH)_4$ utilization was even higher and lower $NO_3^-$ utilization lower than observed today modern values. The calculated Si and N supplies both indicate a slight increase compared to today with Si supply increasing and N supply decreasing towards 14°S and 15°S. The latter agrees is in agreement with continuous denitrification in the southern area causing loss of $NO_3^-$.

In contrast, during the LIA (humid) the decrease in productivity and upwelling has led to more oxygenated waters and subsequent reduction in denitrification devoid of a strong increase in subsurface $\delta^{15}NO_3^-$, which was presumably closer to 6‰ and similar to that of the PCUC reaching the northern shelf area today. Only at 14°S and 15°S the subsurface $\delta^{15}NO_3^-$ signal may have been slightly increased by 1.5‰ due to denitrification if assuming that $NO_3^-$ fractionation in the hydrographic system of the area was continuously consistent with a steady-state type model over time (Fig. 5c).

Generally, AR TN values are by 0.02 (g/cm²/yr) slightly lower during the CWP than today and lowest (<0.02 g/cm²/yr) during the LIA, with little difference between humid and arid phases. The AR BSi values concentrations are generally higher by about 5% 0.1 - 0.35 (g/cm²/yr) during the CWP and LIA (arid) than today with highest values at 15°S, and more similar values to today during the LIA (humid) (Figs. 2, 5). When comparing modern (Fig. 2a) with mean $\delta^{30}Si_{BSi}$ and $\delta^{15}N_{bulk}$ values for the CWP (Fig. 5a) we observe a similar southward increase of both $\delta^{30}Si_{BSi}$ and $\delta^{15}N_{bulk}$ values, with generally slightly higher (0.1-0.2‰) $\delta^{30}Si_{BSi}$ values and with slightly lighter (0.5-1‰) mean $\delta^{15}N_{bulk}$

values than today. A similar trend is observed during the LIA (arid) where $\delta^{15}N_{bulk}$ also reach values of up to 7.9‰ at 15°S, while $\delta^{30}Si$ values are more variable and are even higher (0.4-0.5‰) at 11°S and

12°S than modern values.

During the LIA (humid) $\delta^{30}Si_{BSi}$ values were consistently lower (0.3-0.5‰) than modern values at all latitudes with the only higher values (0.2‰) found at 11°S. The $\delta^{15}N_{bulk}$ values remain close to 5‰, only increasing up to 6.7‰ at 15°S, representing lighter values (0.6-0.3‰) than today, in agreement with the assumption of weakened N-loss processes during this time.

The calculated nutrient utilization shows a shift to high $NO_3^-$ utilization (70-90%) and strongly diminished $Si(OH)_4$ utilization (6-60%; Fig. 5c).

These values are comparable to modern ones from the northern shelf (EQ-10°S), where less nutrients are upwelled and productivity is lower (Ehlert et al., 2012). The calculated $NO_3^-$ supply was lowest during the LIA (humid) with little change over latitude in accordance with prevalence of more oxygenated waters, whereas the Si supply strongly increased especially at 12°S (Fig. 5c). However, the calculated increased $Si(OH)_4$ supply reflects the change in nutrient uptake (i.e. nutrient ratio) due to stratification and potentially Fe limitation rather than an actual increase in $Si(OH)_4$ supply reaching surface waters.

Accordingly we observe a high $Si(OH)_4$ supply but low utilization, thus indicating a low $Si(OH)_4$ demand at the time. In contrast $NO_3^-$ supply appears to be lower than today but the strongly enhanced $NO_3^-$

utilization indicates a higher $NO_3^-$ demand. This shift towards a decreased $Si(OH)_4$ but an increased $NO_3^-$

demand further supports a change in the nutrient uptake ratio by phytoplankton ($NO_3^-$ : $Si(OH)_4$ = 2:1 or

15:1, Fig. 4d). This is in agreement with observations from modern El-Niño events, which show a shift in surface productivity from siliceous (diatoms) to non-siliceous (dinoflagellates) phytoplankton (Sanchez et al., 2000).

Overall we find that the CWP, characterized by high upwelling intensity, productivity and N-loss processes (Fleury et al., 2015; Salvatteci et al., 2014b; Sifeddine et al., 2008), is associated with southward increasing $\delta^{30}Si_{BSi}$ and $\delta^{15}N_{bulk}$ values, reflecting moderate $NO_3^-$ utilization and moderate to high $Si(OH)_4$ utilization (Fig. 6, left). Highest $\delta^{30}Si_{BSi}$ and utilization values at 15°S are potentially caused by progressive Fe limitation during diatom blooms, causing a $NO_3^-$:$Si(OH)_4$ ratio of up to 1:2. Southward increasing $\delta^{15}N_{bulk}$ values and calculated $\delta^{15}NO_3^-$ demonstrate the consistent incorporation of higher isotopic compositions due to subsurface denitrification under anoxic subsurface conditions in agreement with decreasing $NO_3^-$ supply illustrating the N-loss process. Similar conditions prevailed during the sporadic periods of arid conditions observed during the LIA.

In contrast, most of the LIA i.e., the humid phases, was characterized by low productivity and weak denitrification intensity between 10°S and 15°S (Díaz-Ochoa et al., 2009; Salvatteci et al., 2014b;

Sifeddine et al., 2008), with no significant southward increase in the source value of $\delta^{15}NO_3^-$ (Fig. 6, right). Accordingly, high $\delta^{15}N_{bulk}$ and little change in $NO_3^-$ supply indicate more complete $NO_3^-$

utilization during the LIA, while $\delta^{30}Si_{BSi}$ signatures and utilization remained low and Si supply high. This indicates a shift towards a dominance of non-siliceous phytoplankton productivity causing $NO_3^-$

limitation and low uptake of Si similar to observations during modern El-Niño events, which are characterized by a deepening of the thermocline and thus decreased nutrient delivery to surface waters.

**Conclusions**

[revised manuscript text omitted]

Demarest, M.S., Brzezinski, M.A. and Beucher, C., 2009. Fractionation of silicon isotopes during
biogenic silica dissolution. Geochimica et Cosmochimica Acta, 73: 5572-5583.

[revised manuscript text omitted]

Sutton, J., Varela, D., Brzezinski, M.A. and Beucher, C., 2013. Species dependent silicon isotope
fractionation by marine diatoms. Geochimica et Cosmochimica Acta, 104: 300-309.

Takeda, S.: Influence of iron availability on nutrient consumption ratio of diatoms in oceanic waters,
Nature, 393(6687), 774–777, 1998.

Toggweiler, J. R., Dixon, K. and Broecker, W. S.: The Peru Upwelling and the Ventilation of the
South-Pacific Thermocline, J. Geophys. Res., 96(C11), 20467–20497, 1991.

Varela, D. E., Pride, C. J., & Brzezinski, M. A. Biological fractionation of silicon isotopes in Southern
Ocean surface waters. Global Biogeochemical Cycles, 18(GB1047).
http://doi.org/10.1029/2003GB002140, 2004.

Wada, E. and Hattori, A.: Nitrogen isotope effects in the assimilation of inorganic nitrogenous
compounds by marine diatoms, Geomicrobiology Journal, 1(1), 85–101,
doi:10.1080/01490457809377725, 1978.

Waser, N., Harrison, P. J., Nielsen, B., Calvert, S. E. and Turpin, D. H.: Nitrogen isotope fractionation
during the uptake and assimilation of nitrate, nitrite, ammonium, and urea by a marine diatom, Limnol.

Oceangr., 43(2), 215–224, 1998.

Wetzel, F., de Souza, G. and Reynolds, B.: What controls silicon isotope fractionation during
dissolution of diatom opal? Geochimica et Cosmochimica Acta, 131: 128-137, 2014.

Wilken, S., Hoffmann, B., Hersch, N., Kirchgessner, N., Dieluweit, S., Rubner, W., Hoffmann, L. J.,
Merkel, R. and Peeken, I.: Diatom frustules show increased mechanical strength and altered valve
morphology under iron limitation, Limnol. Oceangr., 56(4), 1399–1410,
doi:10.4319/lo.2011.56.4.1399, 2011.

Zuta, S. and Guillén, O.: Oceanografía de las aguas costeras del Perú, Bo. Inst. Mar. Perú, 2(5), 157–
324, 1970.

**Table 1: Downcore record of core M77/2-024-5TC, M77/2-005-3TC and M77/2-003-2TC for $\delta^{30}Si_{BSi}$ (‰) and**
**BSi content (wt%). The 2 SD represents the external reproducibilities of repeated sample measurements.**

| Core | Age yrs BP | Depth (mm) | BSi (wt%) | $\delta^{30}Si_{BSi}$ (‰) | 2SD |
|---|---|---|---|---|---|
| 24-5TC | 42 | 0 | 16.2 | 1.50 | 0.23 |
| | 101 | 42 | 16.1 | 1.26 | 0.17 |
| | 154 | 104 | 34.3 | 1.50 | 0.18 |
| | 170 | 134 | 29.3 | 1.43 | 0.15 |
| | 187 | 161 | 23.7 | 1.47 | 0.05 |
| | 243 | 213 | 30.7 | 1.35 | 0.21 |
| | 304 | 264 | 28.1 | 1.40 | 0.09 |
| | 376 | 301 | 21.0 | 1.38 | 0.16 |
| | 422 | 390 | 10.1 | 0.81 | 0.19 |
| | 441 | 432 | 24.6 | 1.51 | 0.16 |
| | 483 | 473 | 23.8 | 1.61 | 0.08 |
| 005-3TC | 46 | 0 | 15.9 | 1.07 | 0.09 |
| | 73 | 35 | 15.0 | 1.37 | 0.11 |
| | 95 | 69 | 25.4 | 1.46 | 0.21 |
| | 217 | 128 | 18.8 | 1.03 | 0.18 |
| | 250 | 165 | 17.3 | 0.80 | 0.22 |
| | 259 | 185 | 15.1 | 0.93 | 0.13 |
| | 303 | 241 | 13.1 | 0.44 | 0.27 |
| | 340 | 296 | 14.0 | 0.50 | 0.15 |
| | 358 | 323 | 11.6 | 0.47 | 0.20 |
| | 450 | 369 | 14.5 | 1.24 | 0.24 |
| | 464 | 389 | 25.0 | 1.60 | 0.19 |
| 003-2TC | 22 | 0 | 39.2 | 1.63 | 0.24 |
| | 146 | 97 | 40.5 | 1.48 | 0.05 |
| | 245 | 174 | 41.9 | 1.30 | 0.26 |
| | 288 | 208 | 20.8 | 0.65 | 0.23 |
| | 327 | 239 | 23.9 | 0.74 | 0.13 |
| | 411 | 304 | 19.4 | 0.73 | 0.27 |
| | 474 | 353 | 46.7 | 1.38 | 0.17 |
| | 581 | 437 | 29.1 | 0.63 | 0.12 |

---

## Referee Report (RR1)

**Second Review of Doering et al. for BGD**

**By Patrick Rafter\**

**Summary**

First, I'd like to apologize for the tone of my last review—it reads as a more negative review than I intended. This is an improved manuscript and because I better understand the arguments and assumptions being made, I now have more comments to improve the manuscript. First, I would like to see more elaboration on the origins of Fig. 2, especially since it includes new information / calculations. But most importantly the Discussion section needs a complete overhaul.

As I stated in my 1st review, "I don't think the Discussion section is the location for describing every individual wiggle of the observations... No one wants to read a listing of which way the wiggles are wiggling and when. :)" Unfortunately, the new Discussion section is not significantly changed from the earlier manuscript. I think that much of the current Discussion text can be moved or in many cases removed entirely—it is unnecessary to describe every single wiggle and how they relate to every other wiggle. General statements can be useful and the reader can look at the data themselves. Statistics are even better!

To state this differently, the data and the application of the data is interesting and worthwhile, but the Discussion of the data can be much improved by discussing the results in the Discussion section. First, I would categorize most of the Discussion as unnecessary and / or Results section related text. Second, I would suggest how the interpretation of these results is consistent with theories and previous datasets about changes in ENSO variability over the last 600 years. I suggest a complete rewrite of this section with an emphasis on: (1) why these changes are consistent with ENSO and (2) the consistency of the implied changes in ENSO with other datasets.

Line by line notes:

Line 42: result

47: remove "\"

48: confusing "material from the and..."

62: Shouldn't this isotope effect or fractionation factor be negative if the other isotope effects (for uptake) are negative?

91: wrong tilde

118: this assumption of the depth of upwelled water is somewhat arbitrary, but I think it is ok. You could reference a study that has identified the depth of source waters.

121: It is here in the description of Fig. 2A that I realize that how this figure was made has not been described. Am I wrong in thinking that it uses the new data first shown in this manuscript? If so, it seems like the new data should not be included in the Introduction.

144: At this point I again realize that these figures (Fig. 2A, 2B, and 2C) seemingly are using new data that has yet to be introduced. Furthermore, while I think it is important for the reader to understand this spatial variability, the methods used to create these figures (even if they are from earlier work) should be described.

147: The anaerobic oxidation of ammonia (Anammox) does not directly influence the concentration or isotopic value of nitrate.

164-171: This is a good section.

Section 2.5: I had a difficult time understanding this section and it was the second time I have reviewed this technique. I don't have a specific suggestion for editing this section, but I think the authors should take my difficulty into consideration. For example, could this be more easily explained using an illustration? Or an analogy? I'm simply suggesting that they should consider alternate approach for describing their methods here.

265: there is no Fig. 4E

309: Doesn't this sentence need a callout to Fig. 5? The text confuses me because it is seemingly using the MAR as a proxy for upwelling strength (Fig. 3), but the text already quantifies nutrient supply in Fig. 5. These need to be considered together or the text should only use the Fig. 5 estimates.

317: needs a period instead of;

317-318: This last part of the sentence is vague. "In phase" is another statistical term that should not be used to describe wiggles that look like they are going up and down at the same time. Statistics can prove me wrong.

328: this sentence is confusing

330: remove lower

336: this is a great introductory paragraph for the Introduction! Gives the reader a good motivation for why the study is worthwhile. The beginning of the Discussion should be used to restate the question being addressed and point to the Results that

improve our understanding. Also, I am of the mind that time should move forward in the narrative of describing a time series. Beginning with the most recent events and moving back in time is awkward.

344: "latter" refers to the sentence above, but can also be misinterpreted / misread as "later" and should be removed.

348: "This was inferred to result in" is awkward. Reword.

355: element

373: more reference to upwelling strength in Fig. 3. The estimate of nutrient supply rate is one of the cool, new things provided by this study. This should be the focus. Furthermore, it should be made clearer in the manuscript whether the new nutrient supply rate estimates were consistent with he sediment MAR. That is a new contribution to the field.

389-390: Once again, the MAR in Fig. 3 are being used to describe a variety of processes that were (presumably better) estimated later in the manuscript in Fig. 5.

393: I though "correlation" was removed from the text?!!

401: There are a couple instances where the coretop and CWP values are used to estimate nitrate and silicate utilization at the surface. But how well do these compare with the observed modern values?

416: Almost all of this is a boring description of how the different proxies or metrics vary, which is not even altogether necessary in a Results section. It certainly does no good in a Discussion section.

417: this reads to me like equating changes in source nitrate d15N with changes in nitrate utilization. It could happen, but need not be related.

424-426: Despite there being no measurements, the reduction in denitrification predicts a lowering of nitrate d15N. Why can't we use this assumption here?

454: not "changes-in"

460: Another prime example (out of many) of text that belongs in a Results section.

470: Another prime example (out of many) of text that belongs in a Results section.

While it is interesting that these results are mostly consistent with other sedimentary proxy results indicating higher or lower denitrification rates, productivity, etc., the reason we care about this is because of ENSO, right? How does this new data fit within the abundant datasets on ENSO activity over these timescales? I would think that this would be the prime focus of the Discussion section.

\*Why I am signing all my reviews

Full transparency of peer reviews makes reviewers accountable for their work. I say this based on my own experience; my signed reviews are more thoughtful and useful, which leads to better science.

---

## Author Response (AR2)

**Response to the reviewer comments**

Please find in the following a point-to-point response to the reviewers' comments and the corresponding relevant changes in the manuscript.
All major changes in the text are marked in blue in the revised manuscript version.

**Review #1**

My only very minor comment is about the new paragraph on line 304-311. I found this paragraph a little confusing. Firstly, I think the authors meant "The diatom assemblages… show a strong association with…", rather than "strong association of…". Secondly, if there is a strong association between these parameters, then it's a little confusing that in the next sentence that they go on to say that not every change in d30SibSi "is mirrored by a change in diatom assemblage data and vice versa". I think it would be more appropriate to remove the word "strong" from the first sentence, to reduce the risk of misleading the reader.

The sentence has been changed according to the reviewer's suggestion and the word "strong" has been removed.

**Review #2 (Patrick Rafter)**

Summary
This is an improved manuscript and because I better understand the arguments and assumptions being made, I now have more comments to improve the manuscript.

First, I would like to see more elaboration on the origins of Fig. 2, especially since it includes new information calculations.

Regarding the confusion about in fig. 2: All data have been published already (as referenced throughout the text and as now highlighted in the figure caption). It is in fact only the calculations of the accumulation rates for BSi and TN that are new, which has now been emphasized more in the revised version of the manuscript.
Lines 156-174.

But most importantly the Discussion section needs a complete overhaul. Unfortunately, the new Discussion section is not significantly changed from the earlier manuscript. I think that much of the current Discussion text can be moved or in many cases removed entirely. To state this differently, the data and the application of the data is interesting and worthwhile, but the Discussion of the data can be much improved by discussing the results in the Discussion section. First, I would categorize most of the Discussion as unnecessary and / or Results section related text. Second, I would suggest how the interpretation of these results is consistent with theories and previous datasets about changes in ENSO variability over the last 600 years. I suggest a complete rewrite of this section with an emphasis on: (1) why these changes are consistent with ENSO and (2) the consistency of the implied changes in ENSO with other

We agree that the discussion of the changes in the source signatures, nutrient utilization and supply has indeed in parts been too descriptive and has now been significantly shortened. The description of "every wiggle" has been removed and only the important new observations are now discussed.

Regarding the second mayor point of the reviewer we added a new second sub-section to the discussion, in which we discuss our main findings in the context of past ENSO dynamics

determined off Peru and beyond. However, we were not able to follow the suggestion to discuss how the consistency with changes in ENSO compares with other datasets as there are no other datasets describing nutrient utilization or source signature changes. Instead we now point out that this is a new finding.

Line by line notes:

Line 47: remove "\"
"\" has been removed accordingly.

Line 48: confusing "material from the and…"
The sentence was changed to "..., investigations from the water column of the Southern Ocean did not find significant difference between the $\delta^{30}$Si values of particles in the water column and in surface sediments..."

Line 62: Shouldn't this isotope effect or fractionation factor be negative if the other isotope effects (for uptake) are negative?
The reviewer is correct, we added a (-) accordingly.

Line 91: wrong tilde
The symbol was changed accordingly.

Line 118: this assumption of the depth of upwelled water is somewhat arbitrary, but I think it is ok. You could reference a study that has identified the depth of source waters.
It is known that upwelled waters at the Peruvian margin are mainly contributed by the PCUC which prevails at depth of 50-150 m. This is stated in the first sentence of the paragraph: "Along the Peruvian margin the main source for the high amounts of upwelled nutrients (30 µmol L$^{-1}$ for both Si(OH)$_4$ and NO$_3^-$; Bruland et al., 2005) is the subsurface Peru-Chile Undercurrent (PCUC) flowing at depths of 50-150 m,..."

Line 121: It is here in the description of Fig. 2A that I realize that how this figure was made has not been described. Am I wrong in thinking that it uses the new data first shown in this manuscript? If so, it seems like the new data should not be included in the Introduction.
No, indeed all data, except the calculation of AR values for BSi and N, have been published before. This should be evident by the references in the text. A comment about the modification of previous figures has nevertheless been added to the figure caption.

Line 144: At this point I again realize that these figures (Fig. 2A, 2B, and 2C) seemingly are using new data that has yet to be introduced. Furthermore, while I think it is important for the reader to understand this spatial variability, the methods used to create these figures (even if they are from earlier work) should be described.
See comment to line 121. In addition, the calculations used to arrive at the values used in the figure are the same as for the downcore values described in the methods sections. We added a short comment corresponding to this fact: "The here calculated nutrient utilization for surface sediments is identical to the original publications (Fig. 2b; Mollier-Vogel et al., 2012; Ehlert et al., 2012)".

Line 147: The anaerobic oxidation of ammonia (Anammox) does not directly influence the concentration or isotopic value of nitrate.

"anammox" was removed from the sentence.

Section 2.5: I had a difficult time understanding this section and it was the second time I have reviewed this technique. I don't have a specific suggestion for editing this section, but I think the authors should take my difficulty into consideration. For example, could this be more easily explained using an illustration? Or an analogy? I'm simply suggesting that they should consider alternate approach for describing their methods here.

In response to these suggestions we added an illustration including the modern water column surface sediment data by modifying a scheme from Grasse et al. (2016). This illustration is showing modern values from 10°S which are used as an example within the nutrient utilization scheme. This scheme has now been altered to make it easier to understand. These are combined in the new Figure 3a-b. Furthermore, the text in section 2.5 was slightly altered corresponding to the adjustment to the figure.

Line 265: there is no Fig. 4E

The reference was changed to Fig 5b

Line 309: Doesn't this sentence need a callout to Fig. 5? The text confuses me because it is seemingly using the MAR as a proxy for upwelling strength (Fig. 3), but the text already quantifies nutrient supply in Fig. 5. These need to be considered together or the text should only use the Fig. 5 estimates.

We are not sure which paragraph the reviewer is referring to, as the text in line 309 discusses diatom species abundance in relation to upwelling strength. This paragraph has been added in response to a request of reviewer 1 to the previous version of the manuscript. As the mean diatom assemblages shifts are now shown in figure 6 as well, an additional reference to this figure has been added. However, there is still a reference to figure 4 (previous fig. 3), given that it is shows the detailed variation of the records and not only mean value calculations.

Line 317: needs a period instead of ;

";" has been changed to a period.

Line 317-318: This last part of the sentence is vague. "In phase" is another statistical term that should not be used to describe wiggles that look like they are going up and down at the same time. Statistics can prove me wrong.

The sentence has been removed in the new version of the manuscript.

Line 328: this sentence is confusing

The sentence has been rephrased to "The calculated $NO_3^-$ utilization was higher during this time reaching values between 70 and 90%, while $Si(OH)_4$ utilization ranged between 6 and 60%. See line 351.

Line 330: remove lower

"lower" has been removed accordingly

Line 336: this is a great introductory paragraph for the Introduction! Gives the reader a good motivation for why the study is worthwhile. The beginning of the Discussion should be used to restate the question being addressed and point to the Results that improve our understanding. Also,

I am of the mind that time should move forward in the narrative of describing a time series. Beginning with the most recent events and moving back in time is awkward.

The paragraph was been slightly modified and moved to the introduction (now Lines 109-133). Furthermore, the narrative in the overhauled discussion has been restructured, starting with the LIA and moving forward in time.

Line 344: "latter" refers to the sentence above, but can also be misinterpreted / misread as "later" and should be removed.

"latter" has been removed from the sentence

Line 348: "This was inferred to result in" is awkward. Reword.

was reworded to "These changes resulted in "

Line 355: element

the "s" has been removed accordingly

Line 373: more reference to upwelling strength in Fig. 3. The estimate of nutrient supply rate is one of the cool, new things provided by this study. This should be the focus. Furthermore, it should be made clearer in the manuscript whether the new nutrient supply rate estimates were consistent with the sediment MAR. That is a new contribution to the field.

In the overhauled discussion more reference to upwelling strength based on diatom abundance as well as nutrient supply have now been provided. However, the nutrient supply is calculated based on the nutrient utilization and the MAR values. We therefore refrained from discussing the difference between supply and MAR values, as changes in the supply are ultimately dependent on changes in the MAR values.

Line 393: I though "correlation" was removed from the text?!!

Now it is and the word "correlation" has been changed to "comparison"

Line 401: There are a couple instances where the coretop and CWP values are used to estimate nitrate and silicate utilization at the surface. But how well do these compare with the observed modern values?

The core-top values are the modern values we refer to. Of course, there are estimates from the modern water column as well, but these reflect short snapshots of the prevailing conditions, which do not always represent the entire system. However, there is now additional reference to the water column evidence from Grasse et al. (2016) in the new figure 3.

Line 416: Almost all of this is a boring description of how the different proxies or metrics vary, which is not even altogether necessary in a Results section. It certainly does no good in a Discussion section.
Line 417: this reads to me like equating changes in source nitrate d15N with changes in nitrate utilization. It could happen, but need not be related.
Line 460 and 470: Another prime example (out of many) of text that belongs in a Results section.

This paragraphs/sentences have been completely rephrased in the new discussion.

Line 424-426: Despite there being no measurements, the reduction in denitrification predicts a lowering of nitrate d15N. Why can't we use this assumption here?

Of course, the reviewer is right and the $\delta^{15}N$ of nitrate would be lower during El Niño events, as it would be less elevated by denitrification and this is basically what we can show with our data. In the previous version we probably did not sufficiently emphasize this as the basic idea and we now included it in the overhauled discussion

Line 454: not "changes-in"
"-" has been removed